

# Climate influences on sea salt variability at Mount Brown South, East Antarctica

Helen J. Shea[1,2], Ailie Gallant[1,2*], Ariaan Purich[1,3*], and Tessa R. Vance[4]

[1]School of Earth, Atmosphere and Environment, Monash University, Clayton VIC 3800, Australia
[2]ARC Centre of Excellence for Climate Extremes, Monash University, Clayton VIC 3800, Australia
[3]ARC Special Research Initiative for Securing Antarctica's Environmental Future, Monash University, Clayton VIC 3800, Australia
[4]Australian Antarctic Program Partnership, Institute for Marine & Antarctic Studies, University of Tasmania, Battery Point 7004, Australia
*These authors contributed equally to this work.
**Correspondence:** Helen J. Shea (helen.shea1@monash.edu)

**Abstract.** The Mount Brown South (MBS) ice core in East Antarctica (69°S, 86°E) has produced records of sea salt concentration and snow accumulation for examining past climate. In a previous study, the sea salt concentration, but not snow accumulation, showed a significant, positive relationship with the El Niño–Southern Oscillation (ENSO) from June to November. Here, we use observations and reanalysis data to provide insights into the mechanisms modulating this previously identified

relationship for the austral winter season (June–August). A teleconnection between the tropical Pacific and high–latitude winds in the vicinity of MBS is identified. Specifically, El Niño events are related to strengthened westerly winds ∼60°S, leading to more local sea ice via anomalous Ekman transport in an area to the northeast of the MBS site. Impacts from La Niña are less obvious, showing that there is a non–linear component to this relationship. MBS is a wet deposition site, and we show that sea salt is likely transported from northeast of MBS via synoptic–scale storms that accompany high precipitation events. These

storms and their associated precipitation, show no substantial differences between years of high and low sea salt concentration, so we suggest it is the source of sea salt that differs, rather than the transport mechanism. El Niño–associated strengthened westerly winds in the MBS region could enhance sea salt availability by increasing ocean aerosol spray and/or by increasing sea ice formation, both of which can act as sources of sea salt. This may explain why sea salt concentration, rather than snow accumulation, is most closely related to ENSO variability in the ice core record. Identifying the mechanisms modulating key

variables such as sea salts and snow accumulation at ice core sites provides further insights into what these valuable records can decipher about climate variability in the pre–instrumental period.

## 1   Introduction

Reconstructing past climates is useful for understanding climate variability in the context of anthropogenic climate change since the instrumental climate record is relatively short and spatially incomplete. Paleoclimate reconstructions can complement the

observed record through the use of proxies for past climates, which enables context for climate variability and change. This allows for the study of the climate over longer time scales and a wide range of regions, with the aim of potentially constraining



future projections (Dee et al., 2016). The effects of the El Niño Southern Oscillation (ENSO) on West Antarctica have been thoroughly investigated (Harangozo, 2000; Genthon and Cosme, 2003; Turner, 2004; Clem and Fogt, 2013; Etourneau et al., 2013; Clem et al., 2016). However, East Antarctica has comparatively fewer ice core records despite comprising the majority of

ice mass of the continent and thus the influence of ENSO on East Antarctica is less well understood. The Mount Brown South (MBS) ice core (69.111°S, 86.312°E) was drilled in 2017/2018 to fill a spatial gap of data for East Antarctica (Crockart et al., 2021).The climate impacts of ENSO vary across the Antarctic continent; therefore it is important to have a well–dispersed network of ice cores that can be used to understand the long term and region–specific links to climate drivers of Antarctic climate (Li et al., 2021b).


Crockart et al. (2021) found a significant positive correlation between ENSO and the annual sea salt concentrations of the MBS ice core. However, the underlying dynamical processes of the teleconnection pathway of ENSO, and subsequent movement of sea salt to the ice core site is not yet known. MBS is a wet deposition site (Crockart et al., 2021) which means that the ice core constituents are deposited at the site primarily through precipitation. The snow accumulation rate at the MBS site

is recorded in the ice core record and relates to precipitation as well as erosion, ablation and blowing snow (Crockart et al., 2021). Crockart et al. (2021) found the MBS ice core sea salt concentration to have a significant positive relationship with ENSO. However, the MBS ice core annual snow accumulation rate has no relationship with ENSO. Further work is needed to understand why the MBS sea salt concentration and MBS snow accumulation have different relationships with ENSO. Consideration of the link between sea salt and snow accumulation is important since Jackson et al. (2023) found that the MBS ice core

to be biased towards extreme precipitation events (days where daily snowfall amount is in the 90th percentile or higher), with these events representing the largest 10% of all precipitation events but accounting for 52% of the total annual snowfall at MBS.

Sea salt is an important chemical constituent in ice cores, with possible sources including aerosol spray from the open ocean and frost flowers and salty brine from newly formed sea ice (Huang and Jaeglé, 2017; Frey et al., 2020; Thomas et al., 2022).

There is an inverse relationship between sea salt concentrations and temperature throughout seasonal snow layers as well as between interglacial and glacial periods in ice core records (Petit et al., 1981). Both winter months and glacial periods experience higher sea salt concentrations than summer and interglacial periods. The sources of sea salt in Antarctic ice cores are still being debated however the presence or absence of sea ice seems to be important (Severi et al., 2017; Winski et al., 2021). The relationship varies regionally around Antarctica and the processes involved need to be carefully considered.


When analysing the link between ENSO and sea ice, previous studies identify the Antarctic Dipole, which is a quasi–stationary wave in sea ice, sea surface temperature, and surface air temperature (Yuan and Martinson, 2000, 2001; Li et al., 2021a; Lim et al., 2023). It dominates the interannual variability of the sea ice field and the temperature anomalies associated with the Antarctic Dipole produce the largest ENSO signal outside of the tropical Pacific (Liu et al., 2002). The Pacific South American

(PSA) Rossby wave propagation associated with ENSO drives the Antarctic Dipole (Yuan, 2004; Song et al., 2011). During El Niño (La Niña) the Rossby wave creates an arc of low–high–low (high–low–high) pressure anomalies which extends in the





south–easterly direction from the tropical Pacific towards Antarctica (Yiu and Maycock, 2019). The Antarctic Dipole and PSA show there are pathways for ENSO to influence Antarctica.

This study aims to provide mechanistic insights to the relationship between ENSO and the modulation of sea salt variability at MBS in East Antarctica that was identified by Crockart et al. (2021). There are two main aspects of the processes delivering sea salt to MBS that we investigate:

1. The local processes that could transport sea salt to the ice core. Investigation into specific sources of sea salt at MBS is beyond the scope of this study as there is limited information about the production and transfer of chemical markers from the
open ocean and from sea ice to the Antarctic ice sheets (Abram et al., 2013). Rather, here the aim is to understand the local weather and climate that could transport sea salt.

2. The dynamical mechanism of the influence of the tropical Pacific on local processes around MBS. Udy et al. (2021) noted that future work is needed to investigate how the surface conditions are preserved in ice core records and that may assist with linking modes of climate variability with ice cores.

## 2 Methods

## 3 Mount Brown South cite characteristics

The Mount Brown South site is located at 69°S, 86°E and 2,084 m above sea level (Crockart et al., 2021). During different seasons there is only limited variability of prevailing wind speeds and directions shown in model analysis from Vance et al. (2023). Higher precipitation at MBS is linked to easterly winds and lower precipitation is associated with a southerly element
to the winds from the east (Vance et al., 2023). As mentioned above, extreme precipitation events at MBS account for 52% of annual accumulation (Jackson et al., 2023).

### 3.1 Data

To look at the climate events related to the modulation of sea salt concentration at MBS we used the data from the MBS ice core that was presented in Crockart et al. (2021). The log–transformed site average mean annual chloride ($Cl^-$; sea salt
concentration) was detrended from the years 1979–2016 to focus on the observational and reanalysis data period. The data was detrended using linear regression to remove long term trends and highlight the interannual variability. It is important to note that at the time of our analysis, only an annually resolved ice core record was available, therefore all analysis on the sea salt concentrations was annually not seasonally averaged. We elected to consider the annual sea salt concentration similar to Crockart et al. (2021), as the episodic nature of the accumulation regime at MBS makes it difficult to constrain dating to
seasons or parts of the year. However, the annual sea salt concentration is winter dominated (Vance et al., 2024) and can thus be considered a 'polar winter' record.





Sea ice concentration (SIC) from the National Snow and Ice Data Center (NSIDC) Southern Hemisphere version 4 Climate
Data Record was used to investigate the presence or absence of sea ice as a potential source of sea salt. Data are derived from
satellite observations and span 1979–2016. The monthly anomalies were calculated over this period, then the austral winter
mean was taken.

HadISST sea surface temperature (SST) data was used to investigate the El Niño and La Niña signatures (defined in Section
2.2) in the MBS sea salt concentration as well as the tropical influence on MBS local processes. The SST data was re–gridded
to 1° by 1° over the years 1979–2016 to match the SIC data. The monthly anomalies were calculated and detrended using
linear regression, then the austral winter mean was taken. Anomalies in the tropical Pacific specifically related to ENSO tend
to emerge in austral winter (Fogt and Bromwich, 2006).

We use ERA5 at monthly resolution to examine atmospheric circulation. We utilise mean sea level pressure (MSLP) and
800 hPa zonal and meridional winds. The wind pressure level of 800 hPa was used since it is close to the surface but does not
have interference from the land or ocean surface. These variables were used to investigate the extent of the pressure anomalies
propagating from the tropical Pacific and how the circulation around MBS could be influencing the sea ice. The datasets were
re–gridded to 1° by 1° for austral winter over the years 1979–2016. Monthly anomalies were calculated and detrended using
linear regression, then the austral winter mean was taken.


The ERA5 hourly atmospheric reanalysis dataset was used to investigate the possible mechanism for the transport of sea
salt to the ice core on synoptic timescales, as Jackson et al. (2023) found a high proportion of precipitation at the MBS site is
associated with synoptic–scale storms. The reanalysis datasets were re–gridded to 1° by 1° for the austral winters of 1991 and
2014. These two years were chosen as representative samples because they represent the 4th highest (90th percentile; 2014)
and 4th lowest (10th percentile; 1991) years in the MBS sea salt concentration anomalies over 1975–2016. The 90th and 10th
percentiles were chosen because we did not want to analyse the most extreme events in case of unrepresentative outliers. The
variables analysed were the precipitation at MBS (70°S, 86°E), mean sea level pressure, 800 hPa meridional and 800 hPa zonal
wind.

### 3.2  Analysis

Crockart et al. (2021) found a correlation between the sea salt concentration in the MBS ice cores and ENSO during the austral
winter and spring (June – November). They found the multivariate ENSO index to have the strongest relationship with the MBS
ice core, with an r value of 0.533 for sea salt concentration over 1975–2016 (Crockart et al., 2021, Table 1). The multivariate
ENSO index considers the sea level pressure, sea surface temperature, zonal and meridional components of the surface wind,
and outgoing longwave radiation over the tropical Pacific basin (30°S–30°N and 100°E–70°W) (Wolter and Timlin, 2011). We
analysed the austral winter detrended SST anomalies weighted average over different tropical Pacific boxes, and then focused
on the area that had strongest connection with the MBS sea salt concentration anomaly.



A correlation analysis between annual MBS sea salt anomaly and austral winter-averaged Niño4, Niño3.4, Niño3, Niño1+2 SST anomalies (see Table 1 for different ENSO regions latitudes and longitudes); and between the sea salt anomalies and the winter averaged SIC anomalies northeast of MBS was performed over the period 1979–2016. Note that the northeast SIC region was created over 61–64°S, 90–130°E (see Fig. 3), which is an area approximating the SIC anomalies. Correlations were calculated using Pearson's correlation coefficient.

The annual MBS sea salt concentration data anomalies were split into quartiles. The upper quartile included the highest 10 sea salt concentration anomalies, and the lower quartile included the lowest 10 sea salt concentration anomalies (data spanned 38 years from 1979–2016). The Niño3.4 box weighted average (5°N–5°S, 120–170°W) austral winter SST anomalies over the same period were also split into quartiles, also containing 10 data points in the upper and lower quartiles. Composites maps of the upper and lower quartiles were made to analyse how variables averaged over these years differ between the high years and low years. Composite maps were chosen to analyse the data, as opposed to correlation or regression maps, since composite analysis allows for non–linearities in relationships to be examined. The variables analysed were the SST, SIC, MSLP, 800 hPa zonal wind and 800 hPa meridional wind anomalies.

Synoptic–scale storms were investigated as the possible transport mechanism for the sea salt arriving at MBS through wet deposition. Previous studies have noted the extreme precipitation events bias in the MBS ice core (Jackson et al., 2023). Therefore, it was necessary to include precipitation when looking at transport mechanisms. The daily averaged precipitation for austral winter 1991 and 2014 over the MBS site were calculated using the hourly precipitation data from ERA5. The maximum five precipitation days for both years respectively were defined as extreme precipitation events, and then composite maps for 1991 and 2014 were created over those five days using the 800hPa meridional and zonal wind hourly data from ERA5. Analysing the circulation over these five days provides an insight into where the winds were travelling from before they arrived at MBS, since back trajectories were beyond the scope of this study.

## 4 Results

Section 3.1 first looks at which ENSO index has the strongest connection to MBS, then we look at the MBS local processes and compare the high sea salt years to the low sea salt years in Section 3.2. From there we extend this analysis to El Niño years and La Niña years and look at the relationship between the MBS composites and ENSO composites in Section 3.3.

### 4.1 Mount Brown South connection with ENSO

To investigate the ENSO influence on the MBS core, we first consider the annual MBS Cl$^-$ concentration (MBS Cl $^-$) and the correlation to different ENSO indices (Table 1). The MBS ice core was not seasonally resolved at the time of this study therefore austral winter SST in the tropical Pacific was compared to the annually resolved site averaged sea salt concentration.





From Table 1, Niño3.4 was shown to have the strongest connection to the MBS sea salt concentration, with an r value of 0.477, which is significant at the 0.2% level. Therefore, Niño3.4 was used for further analysis.

**Table 1.** Pearson's correlation coefficients for the MBS sea salt concentrations against four different box averaged SST ENSO regions, calculated for austral winter over 1979–2016. Nino3.4 had the highest r value therefore it is the region used in the analysis (shown in bold text).

| ENSO Region | Lat,Lon Range | r value | p value |
| --- | --- | --- | --- |
| Niño4 | 5° N–5° S, 160° E–150° W | 0.404 | 0.012 |
| **Niño3.4** | **5° N–5° S, 170–120° W** | **0.477** | **0.002** |
| Niño3 | 5° N–5° S, 150–90° W | 0.416 | 0.009 |
| Niño1+2 | 0–10° S, 90–80° W | 0.289 | 0.091 |


Crockart et al. (2021) found a positive correlation of 0.457 over the years 1975–2016 between the MBS sea salt concentration anomalies and the Niño3.4 index for austral winter and spring (Crockart et al., 2021, Table 1). These results were recreated (Fig. 1) using only austral winter for the Niño3.4 SST and over the years 1979–2016. The correlation of 0.477 in Fig. 1 is similar to the correlation from Crockart et al. (2021). While Crockart et al. (2021) found a relationship between ENSO and

MBS sea salt concentration, they did not find a statistically significant relationship between ENSO and accumulation in the ice core (r value of 0.147 for Niño3.4, Crockart et al., 2021, Table 1). Therefore, we hypothesise that the ENSO influence on the sea salt variability at MBS could be due to the sea salt aerosol source rather than the deposition dynamics.

## 4.2  Mount Brown South regional climatology

To understand the ENSO–MBS Cl⁻ relationship (Table 1), we first examine the regional conditions around MBS that can

influence the modulation of sea salt concentration in the ice core. Past studies have examined the link between sea salt and sea ice conditions, as well as changes in the regional atmospheric circulation (Dixon et al., 2004; Huang and Jaeglé, 2017; Winski et al., 2021). In this section we analyse SIC, MSLP and winds around MBS, to investigate links to sea ice or changes to the atmospheric circulation transporting sea salt aerosols from the open ocean to the MBS site. We first look at the MSLP, along with SST, to further investigate the ENSO signal.


The MSLP composite maps for upper and lower MBS sea salt concentration quartiles show a high–pressure anomaly above MBS in the MBS upper quartile (Fig. 2a; contours) and a strong El Niño signal in the corresponding SST composite (Fig. 2a; shading). In contrast, in the MBS lower quartile there is a low–pressure anomaly towards the northeast of MBS (Fig. 2b; contours) and no substantial La Niña signal in the SST pattern (Fig. 2b; shading). This shows that there is a nonlinear relation-

ship between the MBS sea salt concentrations and ENSO, with an El Niño signal but no La Niña signal detected. In the upper quartile composite, the pressure gradient leads to strengthened westerly wind anomalies off the coast of MBS. While the lower quartile has a low–pressure system which creates easterly wind anomalies further east off the coast of MBS, compared to the upper quartile (Fig. 2).

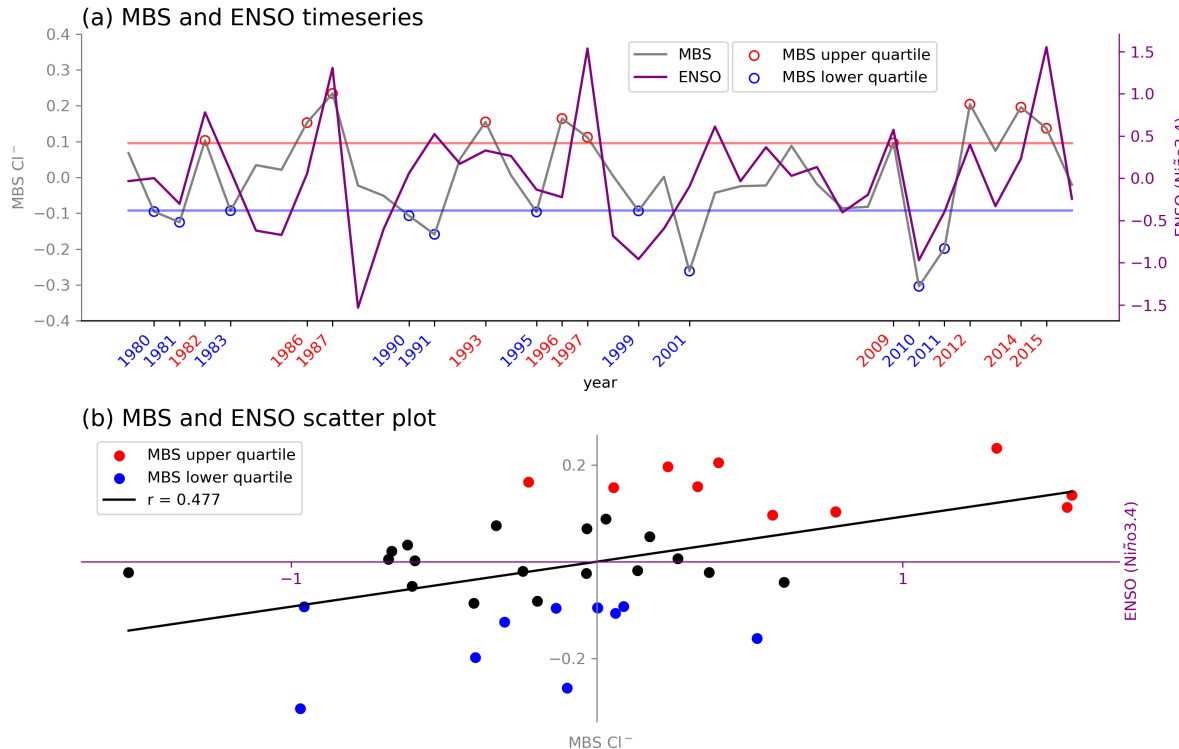

**Figure 1.** a) Time series of the annual detrended, log–transformed sea salt concentrations for the Mount Brown South site average (MBS Cl⁻) and the detrended (June–August) El Niño–Southern Oscillation region SST anomalies (Niño3.4). Red line (blue line) indicates the upper (lower) quartile boundary. b) Scatter plot shows the relationship between the two time series with the MBS upper (lower) quartile highlighted by the red dots (blue dots) based on the period 1979–2016.

Next, the SIC was analysed to investigate the presence or absence of sea ice in the MBS region, as newly formed sea ice is a potential source of sea salt aerosol. North and east of the MBS site, a signal of higher SIC anomalies is observed in the upper quartile composite constructed from the MBS sea salt record (Fig. 3a). Conversely, in the lower quartile composite, lower SIC anomalies are observed (Fig. 3b). This is consistent with previous studies that found higher sea salt concentration during winter and glacial periods when sea ice is more prominent (Petit et al., 1981; Pasteris et al., 2014). Figure 3 shows the high sea

salt years correspond to more prominent sea ice off the northeast coast of MBS. The MBS northeast coast box averaged SIC was chosen to encompass the high SIC anomaly over 61–64°S, 90–130°E (Fig. 3). The correlation between the box-averaged SIC and the MBS sea salt concentration has an r value of 0.448, over the period 1979–2016. This is a statistically significant relationship at the 0.05% level and shows that when there is higher SIC off the northeast coast of MBS, there is higher sea salt concentration in the MBS ice core. This correlation does not give insights into the influence from the tropical Pacific however



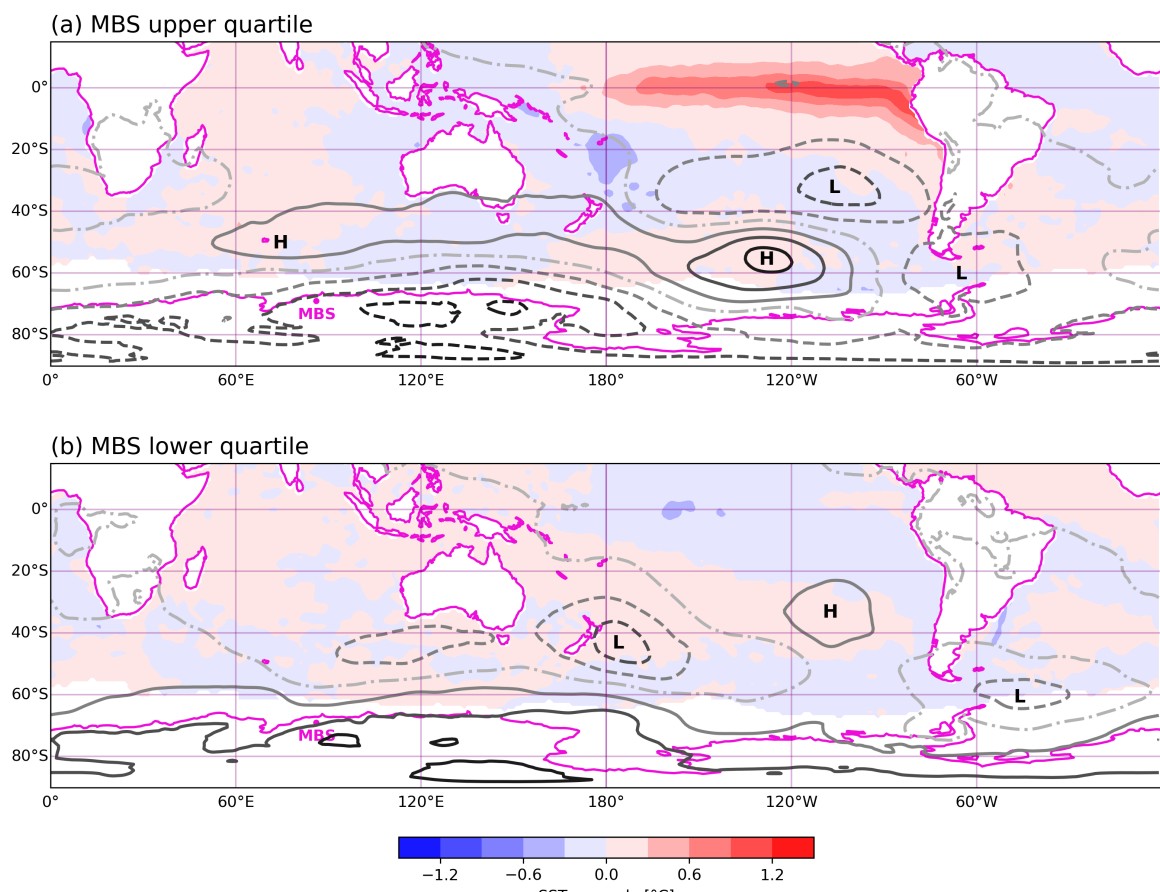

**Figure 2.** Composite maps of the winter sea surface temperature anomaly (shading) and ERA5 mean sea level pressure anomaly (contours) for the Mount Brown South site average (MBS Cl⁻) (a) upper and (b) lower quartile. High (low) MSLP anomalies shown by solid (dashed) contours, with darker shades of grey indicating stronger anomalies. The contour line graduations indicate 0.8 hPa.

this relationship is further explored and compared with ENSO in Section 3.3.

To investigate whether the variance of the source of sea salt is potentially due to the sea ice (or other changes in sea salt aerosol production) or changes to regional circulation patterns associated with the transport of salts, wind anomalies near the MBS region were analysed. Z scores of the meridional and zonal wind anomalies were used, since they are standardized and provide

context as to how strong or weak wind anomalies are with respect to the mean.

The composite maps of 800 hPa wind anomalies showed positive zonal wind anomalies (westerly wind anomalies) along



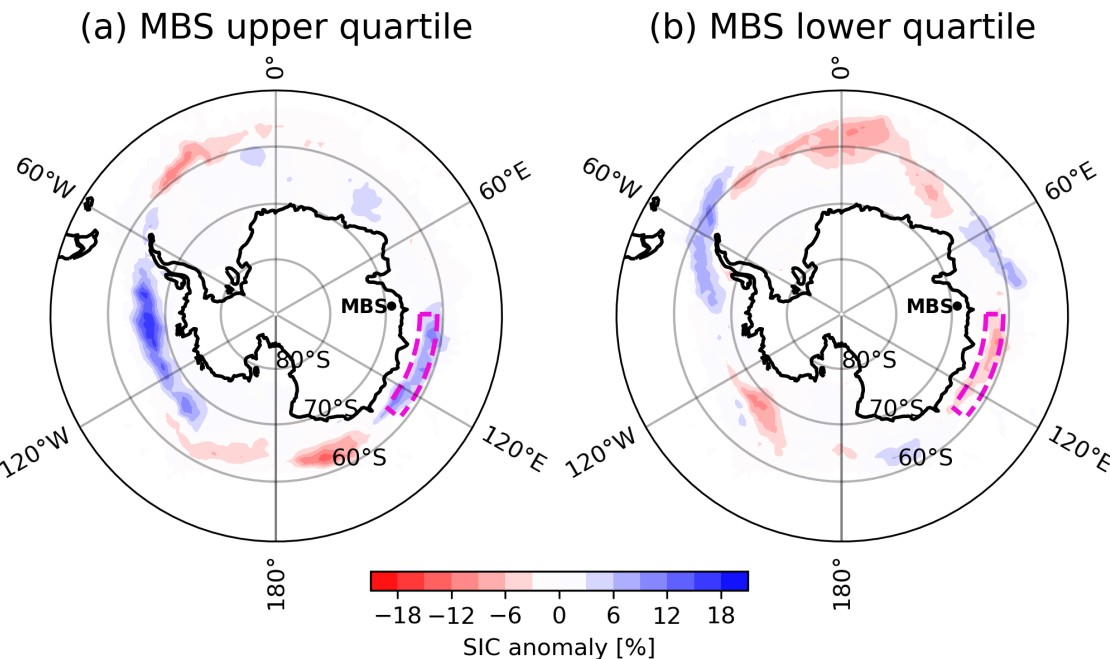

**Figure 3.** Composite maps of the austral winter sea ice concentration for the Mount Brown South site average (MBS Cl$^-$) (a) upper and (b) lower quartile with the MBS northeast coast box (61–64° S, 90–130° E) highlighted by the fuchsia dashed line.

the coast to the east of MBS during years with sea salt concentrations in the upper quartile, and negative zonal wind anomalies (easterly wind anomalies) for the lower quartile (Fig. 4a, c). The direction of these zonal wind anomalies is consistent with anomalous high– and low– mean sea level pressure off the coast of MBS shown in Fig. 2. The composite maps of the meridional wind anomalies showed only small changes in the z scores for years in the upper and lower quartiles of MBS sea salt concentrations. Figure 4 shows small, positive meridional wind anomalies (southerly wind anomalies) around the MBS area with high sea salts, and negative meridional wind anomalies (northerly wind anomalies) for years with low sea salt concentrations.

Changes in both the zonal and meridional wind anomalies for the upper and lower quartiles in the MBS region are less than a standard deviation away from the mean. When Petit et al. (1981) explained how changes to the atmospheric circulation related to high sea salt they suggested the need for stronger winds to transport the sea salt from the open ocean to the ice core site since there is increased sea ice extent during this time. From this it would be inferred the meridional wind anomalies should be stronger to transport the sea salt to the ice core site, however there are no substantial local meridional wind anomalies in either the upper or lower quartiles (Fig. 4b,d). There are, however, stronger zonal wind anomalies for the upper quartile (Fig. 4a) that could relate to increased ocean spray increasing the amount of sea salt aerosol in the atmosphere. The opposite is seen in the





lower quartile (Fig. 4c) where there are weaker zonal wind anomalies that could relate to decreased ocean spray and therefore decreased amounts of sea salt aerosol in the atmosphere. Together, examination of the anomalous zonal and meridional winds in Fig. 4 suggests that the meridional circulation is not the reason for the modulation of MBS sea salt concentrations. Thus, from this evidence we infer that the modulation of sea salt concentration in the MBS core is likely due to a modulation of sea salt variability linked with the anomalous zonal winds. Increased ocean spray and increased sea ice, both linked with strengthened westerly winds, could act as sources of sea salt.

Daily–scale atmospheric circulation was analysed to investigate the possible mechanism of transport of the sea salt to MBS via moisture–laden weather systems. MBS is a wet deposition site (Crockart et al., 2021; Jackson et al., 2023), and the salt is arriving at the site incorporated in precipitation. As described in Section 2.1, the five highest precipitation days from 2014 and 1991 were analysed to represent years of high and low sea salt concentration, respectively, at the MBS site. The circulation over these days provides an insight into the trajectory of the air masses prior to their arrival at MBS. In both years and for all storms examined, the 800 hPa winds were cyclonic over the region of the sea ice anomalies, and the MBS site (Fig. 5). This indicates that, for both high and low sea salt deposition conditions, on the days of highest precipitation the winds blow over the northeast coast box sea ice anomaly area, and then blow towards MBS, depositing sea salt aerosol at the site that has been scavenged by precipitation (i.e. wet deposition). In 2014 there was a positive SIC anomaly to the northeast of MBS, which infers a larger sea ice area. Provided that sea ice is one of the significant sources of sea salt, this shows that there is a larger source of sea salt at this time, which is consistent with the higher sea salt concentration in the ice core. Conversely, in 1991, the SIC anomaly was negative, which is consistent with less sea ice in the region and, subsequently, a lower sea salt concentration in the ice core. The magnitude of the SIC anomalies in both years are notable at around 20–30%. These results support previous studies that have proposed sea ice as a production mechanism for sea salt at Antarctic coastal ice core sites (Thomas et al., 2022). It is also important to acknowledge increased ocean spray as a result of the strengthened westerly winds (Fig. 5a) as another potential source of sea salt at MBS. Our analysis cannot separate these sources of sea salt to MBS. In the real world, it is likely that both sources make contribution to sea salt concentration at MBS. We also note that the year 1991 was characterised by the eruption of Mount Pinatubo, which resulted in global changes in climate during 1991 and following years. We cannot discount that this eruption also impacted both wind variability and SIC in the southern Indian Ocean. Figure 5 shows that even though in Fig. 4 the averaged winter wind anomalies for the upper MBS quartile are blowing from MBS to the coast, processes on higher frequency time scales (i.e. synoptic–scale) may be more important for sea salt deposition at MBS compared to monthly circulation patterns. These findings are consistent with the hypothesis from Jackson et al. (2023) that shows that approximately half of annual snowfall at MBS is from these high precipitation storms.

### 4.3 Insights into the mechanisms connecting the El Niño Southern Oscillation to Mount Brown South

The previous section showed a possible connection between the MBS sea salts and regional sea ice, including the transportation mechanism via synoptic–scale storm systems. Now, the relationships between the El Niño–Southern Oscillation and those localised mechanisms are investigated in order to provide insights into their hemispheric–scale dynamical connections.



**Figure 4.** Composite maps of the averaged austral winter ERA5 800 hPa meridional and zonal wind anomalies for the Mount Brown South site average (MBS Cl$^-$) (a) upper and (c) lower quartile of zonal wind and (b) upper and (d) lower quartile of meridional wind. The MBS northeast coast box is highlighted in fuchsia.

First, the regional circulation around MBS is compared between the MBS upper and lower quartiles and the Niño3.4. The composite maps of circulation anomalies during the upper quartile of MBS sea salt concentration (Fig. 6a, c) and Niño3.4 quartiles (Fig. 6b, d) show similar wind anomaly vectors for the upper quartiles of each, with both having westerlies to south-westerlies (Fig. 6a, b). Conversely, the lower quartiles of each are less consistent, with the MBS composite showing anomalous



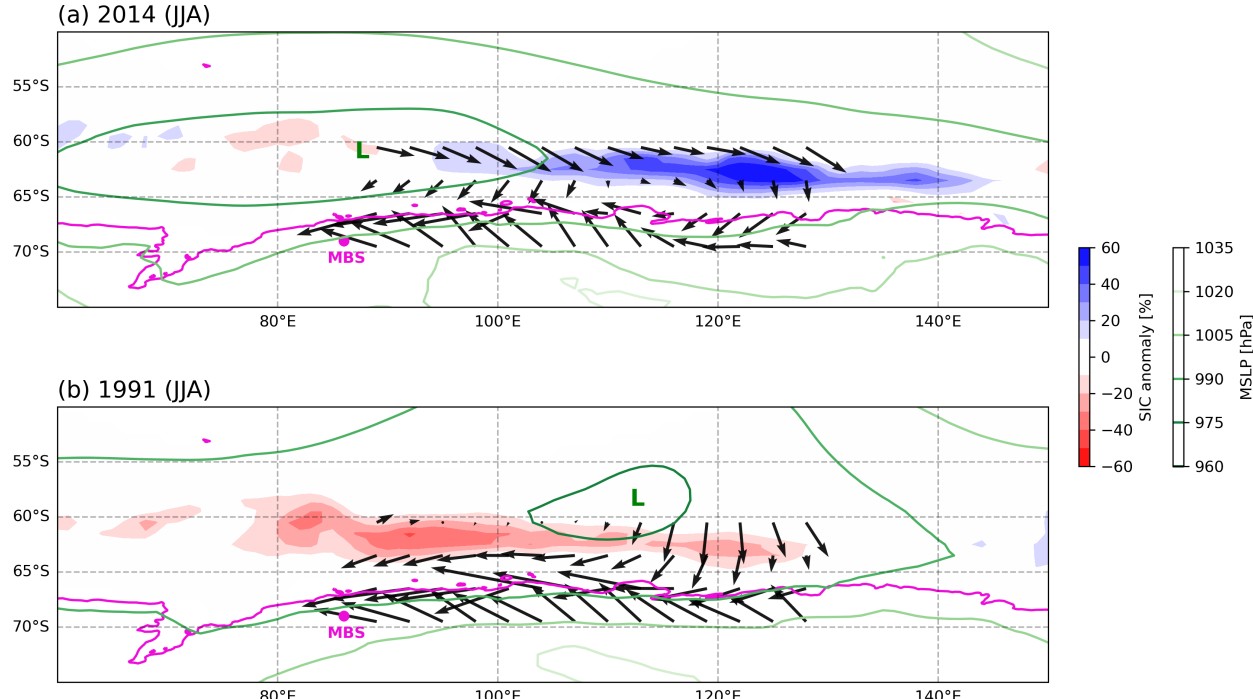

**Figure 5.** Composite maps of the ERA5 mean sea level pressure (contours), ERA5 800 hPa mean wind vectors within (70–58° S, 89–130° W) averaged over the 5 maximum precipitation days in austral winter, and austral winter averaged sea ice concentration (shading) for a) 2014 (90th percentile) and b) 1991 (10th percentile).

easterlies (Fig. 6c), and the Niño3.4 composite showing northeasterlies (Fig. 6d).

The MSLP anomalies for the upper quartiles of MBS and Niño3.4 both show lower pressure anomalies over the land and
coast around MBS, and higher–pressure anomalies north off the coast. As for the wind anomalies, the lower quartiles are less consistent, with higher MSLP anomalies over most of the land and coast in the MBS composite. However, the Niño3.4 composite shows a low–pressure anomaly off the west coast of MBS and a high–pressure anomaly off the east coast of MBS. This is a nonlinear response and is consistent with the SST MBS composites in Fig. 2 that show a nonlinear ENSO relation to MBS. This is, the MBS-ENSO relationship is biased towards influences from El Niño.


The anomalous westerly winds off the MBS coast in the upper quartiles is notable (Fig. 6a, b). As discussed above, this can induce two possible sea salt production mechanisms. Firstly, strengthened westerly winds increase sea salt aerosol lofting from open ocean sea spray. Secondly, in response to the strengthened westerly winds, there will be enhanced northward Ekman transport of sea ice, as Ekman transport theory dictates that sea ice will drift 45° to the left of the wind direction (UCAR, 2008;



Purich et al., 2016). This ice transport mechanism will increase the SIC on the outer (equatorward) side of the ice pack and increase sea ice formation closer to the coast, with newly formed sea ice providing a source of sea salt. The opposite happens during the years with MBS sea salt in the lower quartiles. This is, the easterly wind anomalies tend to reduce open ocean aerosol lofting, as well as reducing northward Ekman transport of sea ice, decreasing the SIC in the outer ice pack, and limiting the formation of new sea ice in the inner ice pack due to lack of exposed ocean there, thus resulting in an overall lower source

of sea salt.

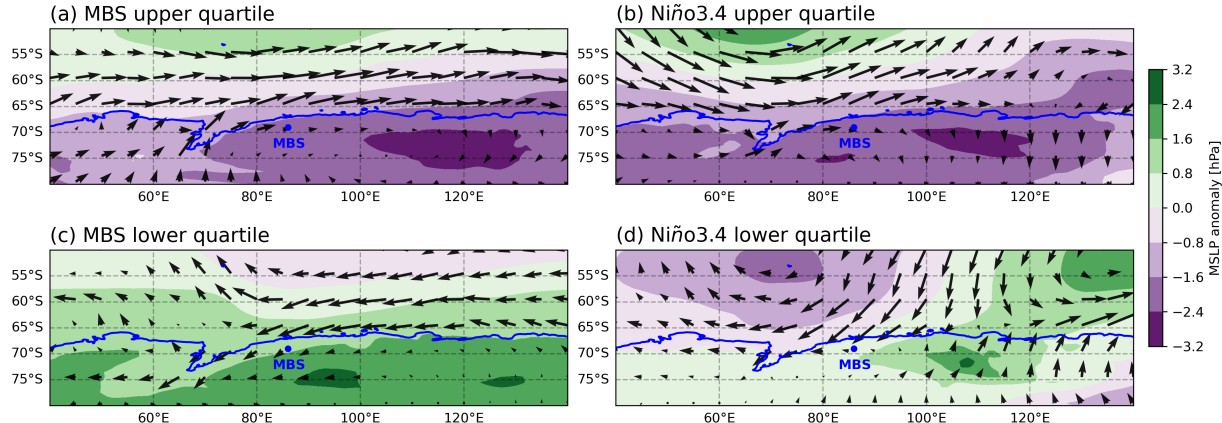

**Figure 6.** Composite maps of the winter ERA5 mean sea level pressure anomaly (shading) with 800hPa wind vector anomalies overlayed for the Mount Brown South site average (MBS Cl⁻) (a) upper and (c) lower quartile and the Niño3.4 (b) upper and (d) lower quartile.

When comparing the SIC in Fig. 3 and Fig. 7, the ENSO signal in the SIC off the northeast coast of MBS is weak (majority of the region in the fuchsia box in Fig. 7). Yet, there is a small ENSO signal at the far northeast corner of the fuchsia box (highlighted as the lime box in Fig. 7) and an El Niño signal in the SST composite pattern for high MBS Cl⁻ years (Fig. 2a).

The lack of a clear ENSO signal in the sea ice directly northeast of MBS could be due to the non-equivalence of SIC (analysed here) and new sea ice formation, which is required for frost flowers, a previously described possible source of sea salt (Rankin et al., 2000).

Another factor for the difference between the Niño3.4 composites and the MBS composites is the location of westerly wind

anomalies. Figure 6 shows that for the MBS upper quartile the westerly winds anomalies stretch to at least 140°E, whereas the Niño3.4 upper quartile westerly winds anomalies become more southerly around 100°E. As described above, the wind anomalies impact the direction of sea ice drift through Ekman transport, and therefore influence where new sea ice forms. The elongated westerly wind anomalies in the MBS upper quartile in Fig. 6 are conducive to the SIC in the MBS upper quartile seen in Fig. 3a.


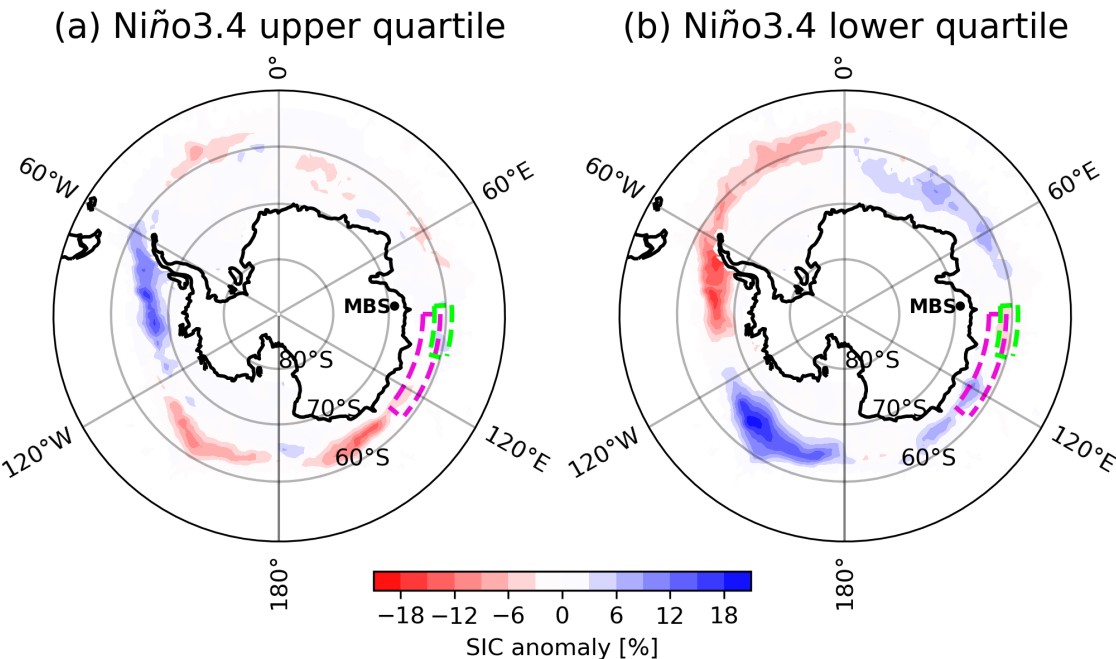

**Figure 7.** Composite maps of the austral winter sea ice concentration for Niño3.4 (a) upper and (b) lower quartile with the MBS northeast coast box highlighted in fuchsia dashed line. Lime box (59–62S,87–105W)

The high MSLP anomaly above MBS in the Niño3.4 upper quartile (i.e. El Niño–like conditions) in Fig. 8a is consistent with the high MSLP anomaly in the same position for the MBS upper quartile in Fig. 2a. The low MSLP anomaly above MBS in the lower quartile of Niño3.4 (Fig. 8b) is not as clearly seen in the MBS lower quartile in Fig. 2b.

The PSA pattern of low–high–low pressure anomalies during El Niño and high–low–high pressure anomalies during La Niña over the southeast Pacific Ocean is seen in both the upper and lower quartiles in Fig. 8. These pressure anomalies are consistent with the known relationship between tropical Pacific SST anomalies, which influences where deep convection occurs in the tropical Pacific (Hoskins and Karoly, 1981), and the propagating Rossby wave trains south and east towards Antarctica (Yiu and Maycock, 2019). When comparing the lower quartiles in Fig. 2 and Fig. 8, the PSA is not as well preserved in the MBS lower quartile compared to the lower quartile of the Niño3.4 (i.e. La Niña conditions). The differences between the lower quartiles in Fig. 2 and Fig. 8 is another indication of the non–linearity between ENSO and MBS sea salt concentration.

A possible explanation for the nonlinear relationship between MBS sea salt concentration and ENSO is that El Niño is related to westerly wind anomalies off the coast of MBS, which allows for more ocean spray aerosol and new sea ice formation,

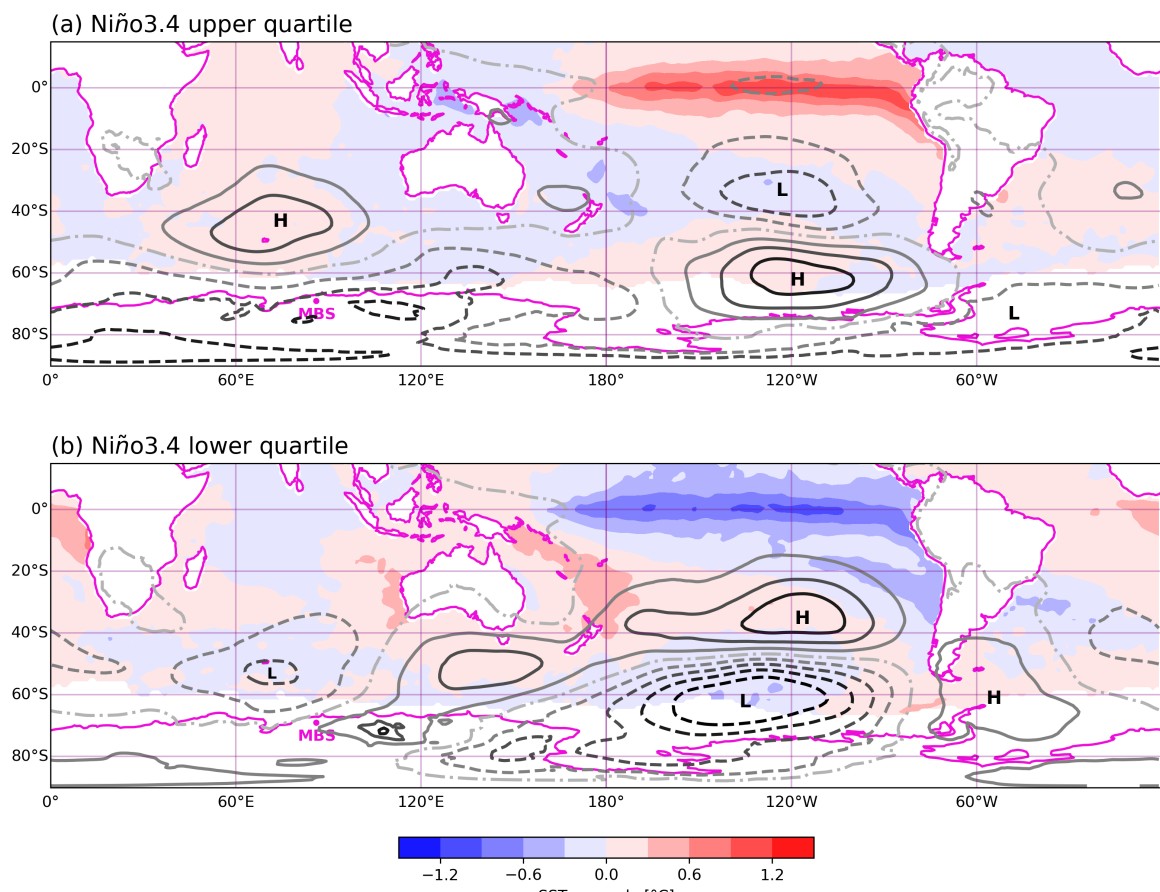

**Figure 8.** Composite maps of the winter sea surface temperature anomaly (shading) and ERA5 mean sea level pressure anomaly (contours) for the Niño3.4 (a) upper and (b) lower quartile. High (low) MSLP anomalies shown by solid (dashed) contours. The contour line graduations indicate 0.8 hPa.

increasing the source of sea salt available to be transported to MBS. However, La Niña may decrease the amount of ocean aerosol spray but does not have a mechanism for decreasing the amount of sea ice formation compared to a neutral state.





## 5 Discussion

Figure 9 provides an overview schematic of the possible localised and remote mechanisms that cause high sea salt concentration in the MBS ice core, based on the evidence we present. The figure shows that, locally, there is some evidence that

high–precipitation, synoptic–scale systems deliver sea salt to MBS, which is consistent with the role of high–precipitation events to the MBS site (Jackson et al., 2023). The area from which the winds in these high–precipitation synoptic systems stem is over a region of high SIC off the coast to the northeast of the MBS site. We hypothesise that the existence of the area of high sea–ice concentration is related to westerly wind anomalies at the coast promoting the production of sea ice, as well as increased ocean aerosol spray, both of which are conducive to higher sea salt concentrations (Frey et al., 2020; Huang and

Jaeglé, 2017). We also suggest that in winter ENSO can influence the winds off the coast of MBS through the known mechanism of the PSA which propagates Rossby waves from the tropical Pacific to Antarctica (Li et al., 2021b). During El Niño the Rossby wave train is more likely to produce a high pressure anomaly off the coast of MBS which creates westerly wind anomalies (Fig. 8). More remotely, this high SIC is weakly related to warm SST anomalies in the tropical Pacific, representing El Niño conditions.


The Law Dome ice core austral summer sea salt concentrations have a negative relationship with ENSO, with high sea salt concentrations associated with a La Niña signature (Vance et al., 2013; Crockart et al., 2021). This is in contrast with our schematic in Fig. 9, showing the region–specific influence of ENSO at MBS. Figure 9 was created using the annual sea salt concentrations from MBS, as opposed to just the austral summer sea salt concentrations used in the Law Dome analysis. The

differences could be due to the different ice core locations and geographies and the wave like pattern of influence on sea ice from ENSO. Udy et al. (2022) found the NDJF high sea salt concentrations in the Law Dome ice core to be associated with increased daily westerly wind anomalies which increase the amount of ocean spray producing sea salt aerosol. The schematic in Fig. 9 shows high sea salt concentration anomalies associated with austral winter averaged monthly westerly wind anomalies off the coast of MBS which is complementary with the findings from Udy et al. (2022). The fact that there are only two ice

cores (MBS and Law Dome) in a large region of East Antarctica, and they have different relationships with ENSO highlights the need for more ice cores in the East Antarctic region, and also highlights the complexity of ENSO-Antarctic climate relationships (Macha et al., 2024).

A previous study of three high accumulation West Antarctic ice cores shows consistencies with the Fig. 9 schematic by sug-

gesting the winter peak in sea salt concentrations are a result of sea ice in the region (Pasteris et al., 2014). Li et al. (2015) was able to reconstruct the sea ice extent in the Ross Sea in early austral winter through sea salt concentration in an interior East Antarctic ice core. They concluded that ENSO influenced the sea ice in the Ross Sea region, and through the transport of sea salt inland, influenced the sea salt deposition at the ice core site. These findings are comparable with our Fig. 9 schematic.




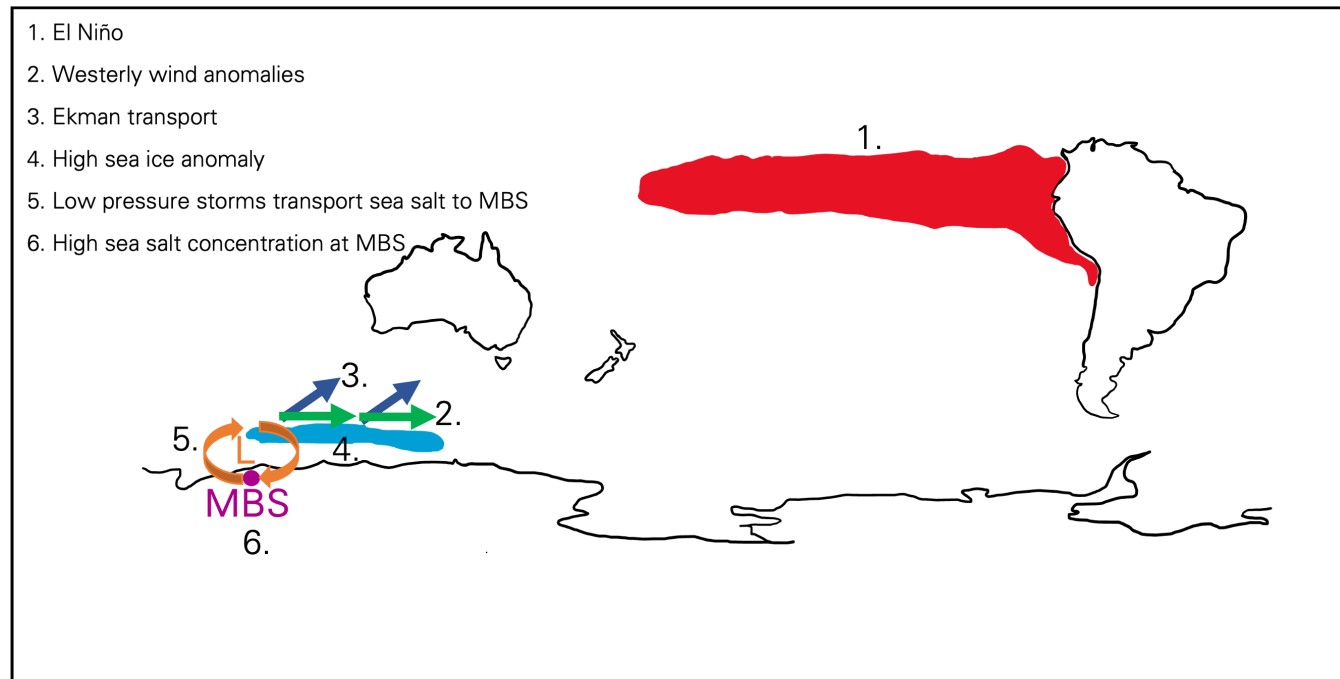

**Figure 9.** Schematic depicting the winter conditions associated with high sea salt concentration in the MBS ice core.





## 5.1 Limitations

We have provided evidence on a possible mechanism for the delivery of sea salt to the MBS ice core. However, the mechanism has been established through evidence from a short observational record, with only 10 El Niño-like and 10 La Niña-like years over the 38–year period. Therefore, it is difficult to separate the influence of ENSO and internal variability unrelated to the tropical Pacific in our analysis. Furthermore, our analysis only looks at the austral winter season. Therefore, the results could be missing other seasonal mechanisms and relationships. Another limitation is around the definition of ENSO that we use – ENSO involves both atmospheric anomalies and ocean anomalies. However, this study only used SST anomalies to calculate the upper and lower quartiles of Niño3.4. The important connection between the tropical Pacific and Antarctica is through the PSA pressure anomaly pattern, which may differ depending on the definition of ENSO. Although ENSO diversity was not explicitly considered here, Table 1 shows that the traditional Niño 3.4 region had the strongest correlations with MBS sea salts. However, the teleconnection to Antarctica also varies between Central and Eastern Pacific El Niño events (Wilson et al., 2014; Macha et al., 2024), which could be an avenue for future research.

Finally, we need to acknowledge that the results of this study show statistical relationships (often weak) of connections between fields, rather than definitive mechanistic pathways influencing the modulation of sea salt concentration in the MBS ice core. Therefore, these results should be considered as insights only. The MSLP composite maps show a potential extension of the PSA towards East Antarctica however the actual mechanism of the Rossby wave propagation from the tropical Pacific was not robustly investigated around the MBS region, as internal variability in circulation patterns overwhelms any coherent signal in the observational data. One avenue that might better interrogate the transport mechanism behind sea salt arriving at the MBS site is back trajectories. However, due to time limitations this was not undertaken in this study (also see Section 4.2).

## 5.2 Implications

The connection between sea salt concentration at MBS and ENSO, and possible linkages to newly formed sea ice off the coast of MBS, is an important insight for using the long ice core record of MBS sea salt as a possible proxy for sea ice. Sea ice is a crucial part of the climate system influencing the surface albedo and interacting with the atmosphere and ocean exchange of heat, moisture, and trace gases such as $CO_2$ (Abram et al., 2013). Regular observations of sea ice began when satellites started collecting data in 1979. Due to this limited sea ice time series, it is difficult to understand how the sensitivity and dynamics of sea ice interacts with the climate system on longer time scales. There is no sea ice observational data for sea ice thickness or age that is useful for recording sea ice formation. Being able to use a climate proxy such as sea salt concentration in ice cores to interpret the low frequency variations in sea ice, in addition to sea ice formation, would be an invaluable resource to look far back in time before observations, and would complement existing ice-core–based reconstructions (Thomas et al., 2019) and station–based sea ice reconstructions (Fogt et al., 2022) of sea ice extent. This would allow longer–term context to be put on the recent, rapid decline in sea ice (Purich and Doddridge, 2023). For example, Li et al. (2015) used the sea salt concentrations from a 2,680 year East Antarctic ice core (79°S, 77°E) as a proxy for sea ice extent in the Ross Sea, finding a significant ENSO

influence. Our study suggests the MBS sea salt concentrations have the potential to provide a new sea ice proxy record for the
last millennium from the full 1,137 year record. However, we also recommend a deeper examination of the sea ice–sea salt
relationship and its causal mechanisms before such an attempt is made.

## 6   Conclusions

This study aimed to understand the mechanisms modulating the relationship between ENSO and sea salt concentration at MBS
in East Antarctica. We showed a relationship between the SIC near the MBS core location, and sea salt from the MBS ice
core. Namely, high sea salt years corresponded to enhanced westerly winds and high SIC off the northeast coast of the MBS
site. Our findings further suggest synoptic–scale storms off the coast of MBS are a possible key transport mechanism for sea
salts from this area of sea ice to MBS. Moving from local processes to the large scale, the previously identified relationship
between MBS sea salt concentration and ENSO (Crockart et al., 2021) was confirmed here. Our analysis suggests that El Niño
events influence local processes around MBS through the generation of Rossby waves via the Pacific South America pattern.
The influence of the PSA extends eastward around western Antarctica towards MBS, creating wind anomalies that impact
both the formation of sea ice and sea salt aerosol from ocean spray. However, the teleconnection pathway between the PSA
and localised circulation anomalies at the MBS site was unclear. Further, this teleconnection pathway appears non–linear, with
connections only evident during El Niño, rather than La Niña events. This highlights the complexity of understanding the
influence of ENSO on Antarctica in the past, and thus the difficulty in projecting how ENSO will interact with climate change
to influence the continent in the future.

*Author contributions.*   T.V. provided the Mount Brown South ice core data; A.G. conceived the study; H.S. analyzed the data with supervision
from A.G. and A.P.; H.S. wrote the manuscript draft; A.G., A.P. and T.V. reviewed and edited the manuscript.

*Competing interests.*   The authors declare that they have no conflict of interest.

*Acknowledgements.*   H.S. and A.G. were supported by the Australian Research Council (ARC) Centre of Excellence for Climate Extremes
(CE170100023). A.P. was supported by the ARC Special Research Initiative for Securing Antarctica's Environmental Future (SR200100005).
T.V. was supported by the Australian Government as part of the Antarctic Science Collaboration Initiative program (ASCI000002). This
research was undertaken with the assistance of resources from the National Computational Infrastructure (NCI Australia) supported by the
Australian government.



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
