# Peer review of "Climate influences on sea salt variability at Mount Brown South, East Antarctica"

_EGUsphere, 2024_

## Author Comment (AC2)

In the response document, editor comments are shown in purple text
In the response document, reviewer comments I have responded to are shown in blue text
In the response document, responses are shown in black text

Line numbers refer to the track changes revised manuscript

You will see that you have comments from three reviewers. They all consider this a useful study, particularly because of the way it opens up the prospect of learning from the 1000 year record that is to come. However they do make important comments about the framing of the paper, and other aspects of the work. Please make sure to respond to all comments when you submit your author comments - only when that has been done can I give a final editorial decision and invite a revised manuscript. From my own viewpoint, I think it is important for you to make clear how this paper adds to what was in the published Crockart paper. From this viewpoint I agree that you may want to emphasise in your framing and abstract a little more the local controls (sea ice, wind patterns) on sea salt, and then the role of ENSO in controlling those as a secondary aspect (this probably requires only minor changes, but is an important distinction because one can imagine the 1000 year record being used as an indicator of past sea ice or local pressure anomalies for example, but i doubt you would suggest we use the long record as an ENSO proxy). Please also clarify the questions about seasonality, which have been raised by two reviewers. I look forward to seeing your responses.

We thank the editor for their time processing our manuscript. We have revised the manuscript in light of the reviewers comments, in particular paying attention to make it clear how our manuscript builds upon the findings of Crockart et al. (2021), distinguishing between local and remote relationships, and discussing seasonality. These are discussed in our point by point response to reviewers below.

**RC1**

The manuscript by Shea et al. presents an interesting interpretation of the mechanisms driving annual sea salt variability at Mount Brown South ice core site (MBS) and their potential relationship with ENSO. The paper is relevant as it provides new insights from a data-sparse region in East Antarctica. The results presented in this paper build upon previous work on MBS published by Crockart et al. (2021), Vance et al. (2024) and Jackson et al. (2023). This manuscript is interesting and relevant and should be considered for publication in Climate of the Past. However, some of the data presented in this study should be revised and/or clarified according to the following considerations, before acceptance for publication in Climate of the Past.

We thank the reviewer for their thorough review of our manuscript. We have addressed their comments (detailed below) and feel that these revisions have improved our manuscript.

Main comments:

(1)     One of the main datasets used in this manuscript is the log-transformed mean annual chloride (Cl-; here considered representative of sea salt concentration). In the absence of higher temporal resolution data, this annually resolved record is used as an austral winter representative. The manuscript highlights that previously, Crockart et al. (2021) used the mean annual sea salt concentration as representative of the June-November period. In this manuscript, the authors take a step beyond and use mean annual sea salt concentrations as June-August representatives. How representative is this, considering that Vance et al. (2024) (Fig 5) show that, during the satellite era, the sodium enhancements reached their maximum during September-October.

This is a good suggestion, which we address below and with new supporting figures in the revised manuscript. We have looked into SON and JJASON with results now in the appendix section. We have determined the regional mechanisms over SON is weaker than JJA therefore focused on JJA in this manuscript. See Appendix Figs A3-A9.

In our initial analysis, we considered the relationships and composite anomalies for both JJA and SON, and finding stronger atmospheric circulation anomalies in JJA, decided to focus on this season. However, you raise a valid point - while Crockart et al. (2021) suggest the annual sea salt concentration represents JJASON, is it reasonable to assume that circulation anomalies in JJA only are causing this? In light of this we have revisited our SON analysis, and include discussion of it here, throughout the manuscript where relevant (lines 195-197 of the track changes revised manuscript) and include additional figures as an Appendix (Figs A3-A9).

Firstly, we revisit the MBS and ENSO timeseries, shown as Fig. 5 in the revised manuscript for JJA and in Fig. A3 for SON and combined JJASON. We include SON panels below as Fig. RR1 for convenience.

[Figure]

*Fig. RR1: Time series of the annual detrended, log--transformed sea salt concentrations for the Mount Brown South site average (MBS Cl-) and the detrended (a) September-November El Niño–Southern Oscillation region SST anomalies (Niño3.4). Red line (blue line) indicates the upper (lower) quartile boundary. Scatter plot shows the relationship between the MBS Cl- and (b) (September-November) Niño3.4 with the MBS upper (lower) quartile highlighted by the red dots (blue dots) based on the period 1979-2016.*

Fig. 5 and RR1 both show a strong positive correlation between the annual MBS timeseries (which we assume to represent the sea salt concentration for JJASON) and the seasonal ENSO index. It is not surprising that these figures are very similar, as the ENSO index between JJA and SON of a given year is very similar.

Next, we revisit the MBS upper and lower quartile composite maps of SST and MSLP anomalies, shown as Fig. 1 in the revised manuscript for JJA and in Fig. A4 for SON and combined JJASON. We include SON panels below as Fig. RR2 for convenience.

[Figure]

*Fig. RR2: Composite maps of the September--November sea surface temperature anomaly (shading) and ERA5 mean sea level pressure anomaly (contours) for the Mount Brown South site average (MBS Cl-) (a) upper [1982, 1986, 1987, 1993, 1996, 1997, 2009, 2012, 2014, 2015] and (b) lower quartile [1980, 1981, 1983, 1990, 1991, 1995, 1999, 2001, 2010, 2011]. High (low) MSLP anomalies shown by solid (dashed) contours, with darker shades of grey indicating stronger anomalies. The contour line graduations indicate 0.8 hPa. The stippling indicates points where the sea surface temperature anomaly (shading) K-S test p value is < 0.1. The fuchsia dot indicates MBS.*

Fig. 1a and Fig. RR2a both show an El Niño-like strong warm anomaly in the eastern equatorial Pacific for the MBS upper quartile years. Fig. RR2b shows a more clear La Niña-like cool anomaly in the eastern equatorial Pacific for MBS lower quartile years during SON then is seen in Fig. 1a for JJA. In both SON, for both upper and lower quartile years, there is a clear teleconnection from the tropical Pacific to the southern high latitudes in the Pacific sector, shown by MSLP anomalies (Fig. RR2). However, interestingly, MSLP anomalies in the MBS region are substantially weaker in SON compared to JJA, particularly in the MBS upper quartile years (i.e. comparing Fig. 1a and Fig. RR2a). These stronger local MSLP anomalies in JJA and weaker MSLP anomalies in SON suggest that the process responsible for the ENSO signal may be occurring predominantly in JJA, associated with the stronger local anomalous circulation. We hypothesise that JJA circulation anomalies in the MBS sector influence MBS sea salt concentrations, while the autocorrelation

between JJA and SON in tropical Pacific conditions results in a continued apparent ENSO influence, despite no substantial circulation anomalies in the MBS region during SON (Lu and Liu, 2019).

Further indication that the MBS sea salt anomalies are driven by processes occurring in JJA come by considering other local climate anomalies for the MBS upper and lower quartile years. Anomalous sea ice concentration (Fig. A6) and 800 hPa zonal wind (Fig. A5) are considerably weaker in SON than JJA, suggestive of unremarkable circulation features during SON. In contrast, regional anomalies in sea ice concentration and zonal wind in JJA (Figs 2, 3 in the revised manuscript) suggest conditions during JJA are more likely to be contributing to variations in the sea salt record. As a result, we continue to present JJA results in our main manuscript, and our focus is on these. However, including discussion of the weaker SON anomalies and reference to the Appendix has been included in the revised manuscript.

(2)    Throughout the text there is no mention of testing the data (MBS sea salt concentration and ENSO) for the presence of outliers which could impact the results obtained from the Pearson's linear correlation coefficient. Figures 1a and 1b show that there are some points which have considerable detours from the mean, potentially biasing the correlation results. An assessment of outliers will make the results more robust.

In response to this comment we have statistically assessed if there are any outliers by undertaking a z score analysis, and testing whether the data used in the Pearson's linear correlation coefficient are not greater than three z score (Kannanet al., 2015). All data points are below this threshold, indicating they are not statistical outliers. In the revised manuscript we have now clarified this. (Lines 195-197 of the track changes revised manuscript).

(3)    Throughout the text it is highlighted that MBS is a "wet deposition site" and that annual precipitation is heavily influenced by extreme precipitation events. However, there is no information about the distribution of extreme precipitation events throughout the year. It is mentioned that "annual sea salt concentration is winter dominated (Vance et al., 2024) and can thus be considered a 'polar winter' record" (Lines 85-86), but there is no further information to know if this is because more extreme precipitation events occurring during winter. Knowing if the extreme precipitation events are bringing the sea salts during winter could contribute to the interpretation of the record.

Thank you for your valuable feedback. We appreciate the opportunity to clarify the points raised.

Wet deposition at the Mount Brown South (MBS) site refers to precipitation primarily arriving from synoptic scale events, such as storms that deposit snow, rather than from clear sky precipitation (diamond dust). The distribution of sea salt throughout the year is based on an assumption of uniform accumulation. Specifically, the amount of snow (in ice equivalent) present between one annual layer horizon and the next is measured, and divide by 12 to obtain 12 'monthly' values. While this method does not account for the variability in daily snowfall, it results in an annual sea salt distribution that is winter dominated, as illustrated by the sodium curve in Figure 5 of Vance et al. (2024) and Figure A3 of Crockart et al. (2021).

Furthermore, surface mass balance and precipitation estimates at the MBS site, as shown in Crockart et al. (2021) Figure 4, indicate a 'polar winter' dominated accumulation regime, with more precipitation occurring between March and November than during December to February. This suggests that the sea salt record may exhibit an even sharper winter peak than the uniform accumulation assumption depicted in Figure 5 of Vance et al. (2024). A winter-dominated sea salt record is common in ice core records, as also observed at Law Dome (Jong et al., 2022).

Understanding the distribution of extreme precipitation events and their contribution to sea salt concentrations at MBS is indeed a significant and intriguing topic. However, it is beyond the scope of this paper, which focuses on the annual mean concentration of sea salt. Future studies could potentially disentangle the different types of events that bring snowfall and their relative salt loads, but this would require extensive research, particularly for periods prior to the satellite era, which is currently not feasible.

We hope this clarifies the points raised and provides a comprehensive understanding of our approach and findings.

(4)    The composite maps presented in Figures 2-8 are a good way of representing the data that is discussed in the manuscript. However, it would be useful to know how consistent these maps are. Do all the years included in the quartiles converge into the same scenarios presented in the composites? Or the composites are biased by some years with a very strong signal/pattern?

This is a good point, thanks for raising it. In the revised manuscript we have analysed the composite years using a K-S test, which determines if two samples come from the same distribution or not. We chose the K-S test given our small sample sizes, as it doesn't require assuming a normal distribution. In this case we compared the upper and lower quartiles, and used a p value of <0.1 to conclude that the upper and lower quartiles were statistically not from the same distribution. This is now shown as stippling on the figures relevant.

(5)    Winds are mentioned as a potential mechanism enhancing sea spray production; however, their potential contribution is always presented relative to the wind anomaly. Previous studies have reported that there is a strong relationship between wind strength and sea spray production, with wind strengths above certain thresholds generating considerably more sea spray. To assess the potential effects of the anomalies over sea spray production, I suggest this manuscript should include a reference to mean wind strength in the region from where sea salts could have been removed. Are the winter regional winds strong enough to produce a considerable enhancement in the MBS sea salt concentration?

Good suggestion - Crockart et al. (2021) analysed the mean winds (their Fig. 9 a,b, copied below as Fig. RR3). In the revised manuscript we have added a comment noting the magnitude of the winds off the coast of MBS. (Lines 99-100 of the track changes revised manuscript).

[Figure]

*Fig. RR3: Figure 9 of Crockart et al. (2021) showing near-surface (10 m) wind vector composite maps of the mean state during austral winter (a, June–August) and spring (b, September–November) and anomaly composite maps during the upper (c) and lower (e) tercile in June–August of the detrended, log-transformed annual sea salt concentrations from the Mount Brown South (MBS) site average, over the period 1979–2016. (d, f) The same as (c) and (e) except in September–November. The red dot indicates the MBS site location. Latitudinal lines are in increments of 5°, and longitudinal lines are in increments of 15°.*

Minor comments:

Lines 22-29: I suggest changing the order of these lines:

"(1)*The effects of the El Niño Southern Oscillation (ENSO) on West Antarctica have been thoroughly investigated (Harangozo, 2000; Genthon and Cosme, 2003; Turner, 2004; Clem and Fogt, 2013; Etourneau et al., 2013; Clem et al., 2016). (2) However, East Antarctica has comparatively fewer ice core records despite comprising the majority of ice mass of the continent and thus the influence of*

*ENSO on East Antarctica is less well understood. (3) The Mount Brown South (MBS) ice core (69.111◦S, 86.312◦E) was drilled in 2017/2018 to fill a spatial gap of data for East Antarctica (Crockart et al.,2021). (4) The climate impacts of ENSO vary across the Antarctic continent; (5) therefore it is important to have a well–dispersed network of ice cores that can be used to understand the long term and region–specific links to climate drivers of Antarctic climate (Li et al., 2021b)."*

I suggest the following order: (4) à (1) à (2) à (5) à (3)

Based on this suggestion, and reviewer 2 and 3 comments, we have removed (1), (2) and (4) and rearranged (5) above (3). (Lines 30-33 of the track changes revised manuscript).

Line 26: consider adding "austral summer" to make it "…was drilled in austral summer 2017/2018 to fill…"

Good point, we have added austral summer. (Line 32 of the track changes revised manuscript).

Line 31: I think the text will be benefit if the R and p values related to the "significant positive correlation" obtained by Crockart et al. (2021) are included here.

Great suggestion, we have added the r and p values in the text here. (Line 37 of the track changes revised manuscript).

Line 31: I suggest specifying that the correlation found by Crockart et al. (2021) is between the June-November ENSO and the annual sea salt concentration, as previously mentioned in the abstract (line 3).

Thanks for the suggestion, we've added this in the manuscript. (Line 37 of the track changes revised manuscript).

Line 36: It seems that this line is repeated (see line 31).

Nice pick up, we have removed this line. (Lines 44-45 of the track changes revised manuscript).

Line 51: please, specify that it is Antarctic sea ice

Thank you for your suggestion; we've updated the manuscript accordingly. (Line 67 of the track changes revised manuscript).

Line 53-54: I believe there is a typo in this sentence which prevents me from understanding the sentence in full. I thought "It" at the beginning of the sentence referred to the Antarctic Dipole, however, it doesn't seem to be the case. It makes sense when removing the words "Antarctic Dipole produce the". I suggest revising this line.

We appreciate your input, and we have made the necessary changes in the manuscript. (Lines 69-70 of the track changes revised manuscript).

Line 60: please consider replacing the word "to" after "insights", for the word "into".

Great point, we've corrected this in the manuscript. (Line 76 of the track changes revised manuscript).

Line 66: please consider adding the word "pattern" or "configuration" after the words "weather and climate".

Great point, we've reworded this sentence in the manuscript. "Rather, here the aim is to relate localised weather and climate circulation patterns to sea salt transport." (Lines 82-83 of the track changes revised manuscript).

Line 71: There is a mismatch with the section number which propagates throughout the text. The subsection currently levelled as 3, should be 2.1. All section numbers and in-text reference to sections have to be re-assessed from this point onwards.

We have fixed the section numbering, thanks

Line 71: change the word "cite" for "site".

Good point, we've corrected this in the manuscript. (Line 89 of the track changes revised manuscript).

Line 72: Please consider including the full decimal coordinates here

Good suggestion, we have added the full decimal coordinates. (Line 90 of the track changes revised manuscript).

Line 90: please specify which months will be used for austral winter estimates (e.g. JJA; JAS; JJAS)

We've incorporated your suggestion and updated the manuscript.

Line 238-241: Figure 5 shows that there is an easterly flow along the coast triggered by the low-pressure field. However, this easterly flow seems to be limited to the coastal region. Wind vectors near MBS seem to be northerly, pushing air from the interior of Antarctica, potentially preventing the access of air from the coast. Could you please comment on this?

Based on the comment of Reviewer 2, in the revised manuscript we now consider all MBS upper and lower quartile years in Fig. 4 in the revised manuscript, to better represent the local circulation features during storms in these two quartiles. In this revised figure, this feature is no longer evident.

Lines 243-301: It will be useful to know which are the years that correspond to the upper and the lower ENSO3.4 quartiles. Maybe a figure in the supplementary, similar to Fig 1 where the upper and lower ENSO quartiles are highlighted. From the scatter plot is clear that MBS and ENSO quartiles are not formed by the same years, but there is no clarity about which years form the ENSO3.4 quartiles

Nice suggestion, we have added this figure to the appendix section (Fig. A2).

Figure 9: I suggest color-coding the numbers in the schematic to better link the graphics with the explanation

Good suggestion, we have added colour coded numbers to Fig. 9.

**RC2**

**Summary**

This paper provides a climatological calibration of the annual sea salt record (log-transformed chloride concentration) from the Mount Brown South (MBS) ice core from 1979-2016. The MBS site is in a data-poor region of the East Antarctic ice sheet in the Indian Ocean sector, with a ~1,000 year-long record that could provide valuable paleoclimate information about sea ice extent, sea level pressure, zonal winds, and even El Nino-Southern Oscillation behavior. In general, I believe this manuscript will be a valuable addition to the literature, assuming that the analysis herein will benefit the interpretation of the longer record. This manuscript also helps to elucidate some of the dynamical controls on sea-salt concentrations in Antarctic ice cores, which is beneficial for the interpretation of other Antarctic ice core sites. However, I have several suggestions for improving the manuscript that I recommend be addressed by the authors prior to publication.

We thank the reviewer for the time and effort in reviewing our manuscript. Below we detail the changes we have made in response to their comments.

**Major Comments**

(1) I think the paper would benefit from a slight re-framing in the abstract and introduction. The hypothesis of Crockart et al. that ENSO significantly influences MBS sea salt concentrations is centered as the motivation from the second sentence of the abstract. However, the title of the manuscript and – in my opinion – the strength of this paper lies with your analysis of the regional sea ice concentration, sea level pressure, and wind drivers of high and low MBS sea salt concentrations. So, I find the ENSO framing early in the paper to be somewhat distracting from the real point of the paper: "what are the controls on MBS sea salt concentration?" While there is potentially an ENSO influence on local sea ice, pressure and winds (as you show), the majority of the variability (~>75%) of those climate metrics is unrelated to ENSO. It seems to me that the strongest motivation for this paper is not that Crockart hypothesized an ENSO connection, but rather that it is a potential ~1000-year record of climate variability from a data-poor region of the continent. The length of the full core is not mentioned until nearly the end of the paper. To me, that's a central part of the motivation for this work.

We appreciate your insightful feedback and have made several adjustments to enhance the clarity and focus of our manuscript. The abstract and introduction have been re-framed to emphasize our analysis of the regional sea ice concentration, sea level pressure, and wind drivers of high and low MBS sea salt concentrations. This shift better aligns with the core strength of our paper, which lies in understanding the local controls on MBS sea salt concentration.

To address your concerns, we have rearranged the methods section to first focus on the MBS regional climatology, providing a detailed context before discussing the ENSO connection. In the results section, we have reordered the subsections to prioritize the MBS regional climatology (previously section 4.2, now section 3.1) and then follow with the MBS connection to ENSO

(previously section 4.1, now section 3.2). This new structure allows readers to first understand the regional climatology before delving into the ENSO-related findings.

Additionally, we have moved the mention of the full core length (1,137 years) to an earlier part of the manuscript to underscore its significance. This change ensures that readers are aware of the extensive temporal scope of our study.

We believe these revisions will provide a clearer and more focused narrative, enhancing the overall impact of our findings.

(2) I think an updated review of the literature on the sources of sea salt in Antarctica would further strengthen the manuscript. In particular, there are many references to the need for "new sea ice" or "sea ice formation" to promote salt emissions, which stems from the focus on salt flowers. Salt flowers were a popular topic in the literature in the early-mid 2000s, but more recent work shows that they may not be as important a sea-salt source as previously thought. There is no mention of blowing snow on sea ice in this paper, despite citing the Huang and Jaegle 2017 paper that emphasizes the importance of that sea salt source. Thinking about wind speed over sea ice – not just new sea ice formation – might lead to a richer interpretation of your zonal wind analysis. The evidence from chemical transport models like GEOSChem, building on the H&J17 work, can help clarify the local partitioning and relative importance of various sea salt sources. While it would be ideal to include something like GEOSChem output in this manuscript to directly inform your interpretations (similar to Winski et al., 2021), it is not absolutely necessary. But including the published work in this area on blowing snow from sea ice, and other recent sea salt source partitioning work, would strengthen this manuscript. One additional figure that would benefit this analysis would be the correlation between SIC within the MBS anomaly region (the Fig. 3 fuchsia box) and 850 mb zonal winds. You'll see a strong positive correlation, with stronger westerly winds associate with higher SIC in that box (I made the figure for my own interest). This figure would support several of your arguments, especially if you add a discussion on blowing snow on sea ice.

Thank you for your valuable suggestion. We have updated the introduction to include recent studies on the sources of sea salt in Antarctica, specifically addressing the role of blowing snow on sea ice as highlighted by Huang and Jaegle (2017). This addition acknowledges the importance of wind speed over sea ice, not just new sea ice formation, in promoting sea salt emissions.

We have also mentioned the use of chemical transport models (GEOS-Chem) in the introduction to clarify the local partitioning and relative importance of various sea salt sources. However, performing such chemical analyses is beyond the scope of our current study.

In response to your suggestion, we have created the figure showing the correlation between SIC within the MBS anomaly region (the Fig. 3 fuchsia box) and 800 hPa zonal winds, and extended this to look at 800hPa meridional winds too (Fig. RR4). This figure supports our findings by demonstrating the strong positive correlation between stronger westerly winds and higher SIC in that region. While this correlation highlights the influence of wind on sea ice concentration, it does not directly address the role of wind in transporting sea salt aerosol to the ice core site. We have revised the manuscript to include the correlation coefficients in the results discussion (lines 255-265 of the track changes revised manuscript). but do not believe it is necessary to also include the figure in the manuscript as well.

We believe these revisions strengthen the manuscript by providing a more comprehensive understanding of the sources of sea salt in Antarctica.

a)

[Figure]

b)

*Fig. RR4: Scatter plot shows the relationship between the SIC within the MBS anomaly region (the Fig. 3 fuchsia box) and 800hPa a) zonal b) meridional winds in the same region for the period 1979-2016. Black line shows the linear regression and r and p values are inserted on the plot.*

(3) All the anomaly figures and related discussion/interpretation would be strengthened if you tested the anomalies for statistical significance. I have concerns that some of the SLP anomalies, for example, are not statistically significant. I think the sea ice and wind anomalies are significant, so I don't think it will change anything major in the take-home results, but it's an essential test to include in all of your figures.

We've taken your suggestion into account and have analysed the composite years using a K-S test which determines if two samples come from the same distribution or not. In this case we looked at the upper and lower quartiles and used a p value of <0.1 to conclude that the upper and lower quartiles were statistically not from the same distribution. This is shown as stippling on the figures relevant.

It is hard to show significance in the MSLP contours (overlain on SST) with stippling in Fig. 1 of the revised manuscript, therefore in the Appendix we included a shaded plot with stippling for MSLP (Fig. A1). This shows the MSLP anomalies to the north of MBS region are not significant, but anomalies over East Antarctica, including the MBS region are significant. We have amended the manuscript to talk about the MSLP anomalies, noting where they are significant. (Lines 206-209 of the track changes revised manuscript).

(4) The synoptic/daily analysis shown in Figure 5 would be strengthened by using more than one low sea salt year (1991) and one high sea salt year (2014), and by using more than five high-precip days in each of those years using ERA5 precipitation data. This is particularly the case because the ERA5 precipitation data is known to have significant biases, especially in data-poor Antarctica (see Roussel, Marie-Laure, et al. "Brief communication: Evaluating Antarctic precipitation in ERA5 and CMIP6 against CloudSat observations." *The Cryosphere* 14.8 (2020): 2715-2727). I'm left wondering if the differences between Figure 5a and b are real and important (e.g. the different positions of the lows), or simply a result of having such a small "n". This is relevant given the potential nonlinear relationships with ENSO discussed herein, and also relevant given the Pinatubo eruption in 1991. As mentioned in the "Limitations" section, that makes 1991 difficult to interpret on its own.

Thank you for your insightful suggestion. In response to your comment, we have revised the manuscript to include all years in the upper (a) and lower (b) quartiles for the synoptic/daily analysis shown in Figure 4 of the revised manuscript. We have also expanded our analysis to include 10 and 20 high-precipitation days, in addition to the highest 5 days, using ERA5 precipitation data (Fig. 4 and Fig. RR5).

Our results indicate that the patterns observed are consistent across these different sample sizes, which suggests that Figure 4 is robust and not merely due to a small sample size. It is recognised that ERA5 tends to overestimate precipitation; however, this overestimation is consistent. Consequently, high precipitation events in ERA5 are likely to correspond to high precipitation events in reality, despite the wet bias in the reanalysis. This consistency ensures that the ranking of the top 5, 10, or 20 precipitation events remains relatively accurate, effectively illustrating our point. This is further supported by the sensitivity analysis of the wettest 10 and 20 events.

We acknowledge the known biases in ERA5 precipitation data, particularly in data-sparse regions like Antarctica, as highlighted by Roussel et al. (2020). However, ERA5 is largely regarded as the best reanalysis for Antarctica (Hassler and Lauer, 2021), and given that we are considering circulation, winds and precipitation together, we need to use a reanalysis product for this. We've added this discussion to the manuscript. (Lines 137-140 of the track changes revised manuscript).

[Figure]

*Fig. RR5: Composite maps of the daily ERA5 mean sea level pressure (contours), daily ERA5 800 hPa mean wind vectors averaged over the (a,b) 10 (c,d) 20 maximum precipitation days in austral winter, and austral winter averaged sea ice concentration (shading) for the Mount Brown South site average (MBS Cl−) (a,c) upper [1982, 1986, 1987, 1993, 1996, 1997, 2009, 2012, 2014, 2015] and (b,d) MBS lower quartile [1980, 1981, 1983, 1990, 1991, 1995, 1999, 2001, 2010, 2011]. The fuchsia dot indicates MBS.*

(5) Given that the dynamical link between the tropical Pacific/ENSO and Antarctica is via atmospheric teleconnections, I think it would be informative to investigate the ENSO indices that include atmospheric components, or even the SOI itself. If correlations between MBS and MEI or BEST or SOI are weaker than those with Nino3.4, then it seems likely to me that the relationship between ENSO and MBS is weaker than it appears using Nino3.4. In other words, stronger r values with Nino3.4 may simply be due to chance as you are correlating two rather noisy, and rather short, time series. I can't think of a dynamical reason why tropical SSTs would have a stronger influence on MBS than the atmospheric anomalies that those SSTs induce, given that the atmospheric anomalies are a step closer to MBS in the hypothesized cause-effect progression.

Thank you for your valuable suggestion. In response to your comment, we have considered the SOI and its potential correlations with MBS sea salt concentrations. Firstly, we note that Crockart et al. (2021) analyzed the SOI and found a significant correlation between SOI and MBS sea salt concentrations for JJASON (their Table 1, excerpt copied below as Table RR1), which is comparable to the other indices analyzed.

We extended this analysis to the JJA period and also found a significant correlation between SOI and MBS (Table RR2), consistent with the other indices. We have added the SOI correlation to Table 1 in the revised manuscript. Additionally, we examined SST and MSLP for the upper and lower quartiles of SOI (Fig. RR6), which showed similar patterns to those observed for the Nino3.4 upper and lower quartiles (Fig. 8).

These findings reinforce our confidence in using the Nino3.4 index to represent the coupled ocean-atmosphere ENSO phenomena in this manuscript.

*Table RR1: Excerpt from Table 1 in Crockart et al. (2021) Pearson's correlation coefficient for the detrended annual log-transformed sea salt concentrations for the Mount Brown South site average (MBSsaltSite) against detrended seasonal ENSO indices. Correlations significant at 95 % are bold, and the dates range between 1975–2016.*

|  |  | Seasonal indices | Range | *r* value | *p* value |
|---|---|---|---|---|---|
| MBSsaltSite | **ENSO (Multivariate ENSO Index)** | **JJASON** | **1979–2016** | **0.533** | **0.001** |
|  | **ENSO (4 Index)** | **JJASON** | **1975–2016** | **0.521** | **0.000** |
|  | **ENSO (3.4 Index)** | **JJASON** | **1975–2016** | **0.457** | **0.002** |
|  | **ENSO (Southern Oscillation Index)** | **JJASON** | **1975–2016** | **–0.496** | **0.001** |

*Table RR2: Pearson's correlation coefficients for the MBS sea salt concentrations against four different box averaged SST ENSO regions and SOI, calculated for austral winter over 1979–2016. Nino3.4 had the highest r value therefore it is the region used in the analysis (shown in bold text).*

| ENSO Region | Lat,Lon Range | r value | p value |
|---|---|---|---|
| Niño4 | 5° N–5° S, 160° E–150° W | 0.404 | 0.012 |
| **Niño3.4** | **5° N–5° S, 170–120° W** | 0.477 | 0.002 |
| Niño3 | 5° N–5° S, 150–90° W | 0.416 | 0.009 |
| Niño1+2 | 0–10° S, 90–80° W | 0.289 | 0.091 |
| SOI | | −0.451 | 0.004 |

(a) SOI lower quartile

(b) SOI upper quartile

SST anomaly [°C]

[Figure]

*Fig. RR6: Composite maps of the (a,b) winter sea surface temperature anomaly (shading) and (a-d) ERA5 mean sea level pressure anomaly (a,b) (contours) (c,d) (shading) for the SOI (b,d) upper [1981, 1988, 1989, 1996, 1998, 2008, 2010, 2011, 2013, 2016] and (a,c) lower quartile [1982, 1987, 1992, 1993, 1994, 1997, 2002, 2004, 2006, 2015]. (a,b) High (low) MSLP anomalies shown by solid (dashed) contours. The contour line graduations indicate 0.8 hPa. The blue dot indicates MBS.*

(6) I strongly suggest the authors use a winter-spring seasonal average rather than a winter average only for the analyses in this manuscript. Other Antarctic ice cores and aerosol monitoring stations show that sea salt deposition peaks through September and even October. This is, likely, related to the importance of sea ice on sea salt deposition combined with the seasonal lag of sea ice. The ideal months to include would perhaps be July-October, but JJASON would also be better than JJA. The fact that Crockart et al. also use a winter-spring seasonality for their analysis of this same MBS core is further motivation to use that seasonality in this paper.

Thanks for this suggestion. RC1 also had a similar suggestion - this is something we have considered carefully. Please refer to our detailed response to RC1 above. To summarise, we have now expanded our analysis to include both SON and JJASON, and include results in the Appendix (Figs A3-A9). We determined the regional mechanisms over SON are weaker than JJA, therefore continue to focus on JJA in this manuscript.

Minor Comments:

The captions for Figures 2, 3, 4, 6, 7, 8 should include the time period (years) represented by the data in each figure.

Great point, we've added this in the manuscript.

Line 28: "climate drivers of Antarctic climate"  delete the first "climate"

Good point, we've corrected this in the manuscript. (Line 31 of the track changes revised manuscript).

Line 33: missing comma after "site"

Thanks for highlighting that, we've adjusted this sentence in the manuscript due to the reframing. The new phrasing is "However, the subsequent movement of sea salt to the ice core site, and the underlying dynamical processes of the teleconnection pathway of ENSO, is not yet known.".(Lines 38-40 of the track changes revised manuscript).

Line 36: Same sentence largely repeated from line 31.

Thank you for your suggestion. We have already made the change as recommended by RC1.

Line 39: delete "that" after "found"

Thanks for pointing that out, we've fixed it in the manuscript. (Line 47 of the track changes revised manuscript).

Line 44: Blowing snow from sea ice should be added to this sentence as an important sea salt source. It's probably more important than frost flowers, and seasonally more important than the open ocean. See the paper you cite by Huang and Jaegle, 2017 (and note the title of their paper).

You've made an excellent point, and we've updated the manuscript to reflect that. (Lines 52-53 of the track changes revised manuscript).

Line 71: Section "3" is really section 2.1, correct?

Yes, this has been corrected.

Lines 72-76: The section on Mount Brown South site characteristics requires more details for the reader. Please add mean annual temperature and mean annual snow accumulation of the site; the accumulation is particularly important for readers to put the repeated references to a "wet deposition site" into context. Also, please explicitly state for the reader that the Mount Brown South ice core "site average" sea salt record discussed here represents the average of three ice cores collected within ~100 m of one another. This is important context for the reader. I would also suggest you state that the three ice core sea salt records correlate with one another with an r value between 0.43 to 0.59, and they each correlate with the site-averaged record with an r value >0.8 (and cite Crockart). Again, this is helpful context for the reader to evaluate your correlation coefficients. Also, please clarify if you are using the raw annual or Gaussian smoothed annual site-averaged sea salt

record from Crockart in this analysis. Also include the time length of the full MBS record here. You mention that it is 1,137 years long only towards the end of the manuscript.

Thank you for your detailed feedback. We agree with your suggestions and have revised the manuscript to include the mean annual temperature and mean annual snow accumulation for the Mount Brown South site. We have also explicitly stated that the Mount Brown South ice core "site average" sea salt record represents the average of three ice cores collected within ~100 m of one another. Additionally, we have included the correlation coefficients. We have clarified that we are using the raw annual site-averaged sea salt record from Crockart in our analysis. Finally, we have included the time length of the full MBS record (1,137 years) in section 2.1 of the manuscript.

Line 83: "annually not seasonally averaged". Perhaps "annually rather than seasonally"?

That's a helpful suggestion, and we've corrected it in the manuscript. (Line 109 of the track changes revised manuscript).

Line 83-85: These consecutive sentences are a bit at odds with one another. The first says you used the annual averaged data because that was all that was available. The next says that you "elected" to use the annual averaged data because of the episodic nature of the snow accumulation. Which is it?

Thanks for the feedback. For clarity, we've removed the sentence 'We elected to consider the annual sea salt concentration similar to Crockart et al. 2021, as the episodic nature of the accumulation regime at MBS makes it difficult to constrain dating to seasons or parts of the year.' from the revised manuscript. (Lines 109-111 of the track changes revised manuscript).

Line 89: "presence or absence of sea ice". Only the presence or absence (e.g., 0 or 1), or the varying concentration of sea ice? I'm sure it's the latter.

Great observation, we've made the necessary correction in the manuscript. (Line 116 of the track changes revised manuscript).

Line 91 (and 96, and elsewhere): Which months constitute the "austral winter mean"? Please be specific about which months are included (I'm assuming JJA), and if that varies at all within this analysis.

Thank you for highlighting that. We have already made the change as recommended by RC1.

Lines 93-95: This SST section should clarify – are you analyzing SSTs in the Southern Ocean or in the tropical Pacific? I think the later (based on the next page), but it should be clarified here.

Thank you for the suggestion. We have added "gridded sea surface temperature (SST) data was used to investigate the local and remote SST anomalies during high and low sea salt years, including El Niño and La Niña signatures in the tropical Pacific." to the manuscript. (Lines 142-143 of the track changes revised manuscript).

Line 93: "Defined in section 2.2" – I don't know what this is referring to, as there is no section 2.2 But this section would be section 2.2 "Data" if it were numbered correctly, I think. Perhaps this means section 2.3 "Analysis"?

Thanks for bringing that to our attention, we've revised the manuscript accordingly.

Line 93: Define the acronym "HadISST" the first time you use it, and it needs a reference.

We appreciate your feedback and have made the necessary corrections in the manuscript. (Line 142 of the track changes revised manuscript).

Line 99: Define the acronym "ERA5" the first time you use it, and it needs a reference.

Thanks for your feedback. We've corrected it in the manuscript. (Line 121 of the track changes revised manuscript).

Line 106: Should be "investigate possible mechanisms"

Thanks for pointing that out, we've fixed it in the manuscript. (Line 130 of the track changes revised manuscript).

Lines 123-125: This sentence should be revised for clarity and flow – semicolon should be a comma, "correlation analysis" should be made plural, and the sentence structure is awkward.

Thanks for highlighting that, we've amended it in the manuscript. (Line 183 of the track changes revised manuscript).

Line 133-134: "high years and low years" should be "highest quartile and lowest quartile". For which geographic region(s) are you creating these composite maps?

We have replaced high and low years with "upper quartile and lower quartile". Composites are calculated using gridded data, focussing on the Southern Hemisphere for SST and MSLP regions, and the southern polar region for sea ice concentration and wind anomalies. (Lines 156-157 of the track changes revised manuscript).

Line 139: "events" should be "event"

You're right, we've updated the manuscript accordingly. (Line 165 of the track changes revised manuscript).

Line 141: should be "was calculated"

That's a valid point, we've corrected it to 'are' in the manuscript to show present tense . (Line 167 of the track changes revised manuscript).

Line 147: Double check the section numbering throughout the manuscript.

Thanks for pointing that out, we've fixed it in the manuscript.

Line 151: Should be "we first consider correlations between the annual MBS Cl- concentration (MBS Cl-) and different ENSO indices (Table 1)"

Good catch, we've made the necessary change in the manuscript. (Lines 294-295 of the track changes revised manuscript).

Line 152: missing comma after "this study".

Thanks for highlighting that, we've added it in the manuscript. (Line 296 of the track changes revised manuscript).

Line 156-157: I'm curious about why you would limit your analysis to the austral winter (presumably JJA, although not defined) given that a) Crockart focused on winter-spring, and more importantly b) sea salt seasonal cycles generally peak in winter-spring, not just the winter. This can be seen in any number of papers on Antarctic sea salt seasonality in both ice cores and aerosol stations; for example, in Winski et al., 2021, cited herein. This winter-spring seasonality reflects the importance of sea ice as a sea salt source.

We have addressed this from your main comment.

Line 167: extra comma. Should be "…winds around MBS to investigate…"

Thanks for highlighting that, we've deleted it in the manuscript. (Line 202 of the track changes revised manuscript).

Line 171: "high-pressure anomaly above MBS in the MBS upper quartile". The word "above" in this sentence is confusing. Do you mean "north of MBS" in the southern Indian Ocean? Or do you mean in the atmosphere on top of the MBS site? I think you mean the former since on top of the MBS site in the upper quartile plot is a low-pressure anomaly.

You've made an excellent point, and we've updated the manuscript to reflect that. (Lines 207 and 209 of the track changes revised manuscript).

Figure 1: Caption should indicate that the red and blue years in panel (a) x axis represent the MBS upper and lower quartile years.

That's a helpful suggestion, and we've corrected it in the manuscript.

Figure 2: It is important to indicate which pressure and SST anomalies are statistically significant given the variability of the respective datasets. I doubt, for example, that the ~0.8 hPa high pressure anomaly north of MBS in Figure 2a is statistically significant. The H pressure anomaly ~55 S, 130 W of Figure 2a probably is significant, however, along with the tropical SST anomalies. I similarly doubt the low-pressure anomaly NE of MBS in Figure 2b is statistically significant.

We have addressed this from your main comment.

Line 180-181: It's not just the "presence or absence" of sea ice, but rather changes in sea ice concentration. Further, sea salt aerosol does not only come from "newly formed sea ice", but from blowing snow on sea ice, as described nicely in Huang and Jaegle, 2017.

We've incorporated your suggestion and updated the manuscript. (Lines 246-247 of the track changes revised manuscript).

Line 184-185: The sentence, "Figure 3 shows the high sea…" is unnecessary, as it repeats info from the prior few sentences.

Thank you for your suggestion; we've updated the manuscript accordingly. (Lines 251-252 of the track changes revised manuscript).

Line 185-186: "The MBS northeast coast box…" sentence is awkward (please revise), and it should mention that this box encompasses not just the positive SIC anomaly but also the negative SIC anomaly.

Thanks for highlighting that, we've addressed it in the manuscript. (Line 252 of the track changes revised manuscript).

Line189: Missing comma. Should be "…topical Pacific, however"

Thanks for the feedback, we've removed this sentence in the manuscript from reframing sections. (Lines 264-265 of the track changes revised manuscript).

Line 191: You don't know that there is a "variance in the source of sea salt", only that there is variability in the MBS sea salt annual concentrations. Separately (grammatically), I recommend moving "wind anomalies near the MBS region were analyzed" to the start of the sentence.

That's a helpful suggestion, and we've corrected it in the manuscript. (Lines 216-220 of the track changes revised manuscript).

Line 196: Check verb tenses throughout. You generally use present tense (which is recommended) in the manuscript, but here and in some other places you write that the maps "showed" things. Should be "show".

Great observation, we've made the necessary corrections in the manuscript.

Line 199-200: I suggest something like, "The direction of these zonal wind anomalies is consistent with the mean sea level pressure anomalies associated with the highest and lowest quartiles shown in Fig. 2."

Your suggestion was very helpful, and we've resolved this in the manuscript. (Lines 226-227 of the track changes revised manuscript).

Lines 238-241: I think the sentence that begins "Figure 5 shows..." is based on an incorrect premise. Note that Figure 4 shows the meridional (and zonal) wind anomalies. They do not show the actual direction of those winds. It is possible (likely?) that, due to katabatics, the meridional winds are

going from MBS towards the coast on a monthly average basis. But certainly during the storms that deposit snow and sea salt on MBS the meridional winds would be onshore as you suggest. But to be clear, Fig. 4 does not show offshore meridional winds – it shows offshore wind *anomalies*. So the text needs to be revised.

Good point, we have added (or weaker winds from the coast to MBS) in the revised manuscript thanks. (Lines 287-288 of the track changes revised manuscript).

Lines 265-270: In this section, make sure it is clear that easterly wind anomalies in the westerly wind belt means (on a monthly basis) weaker westerly winds, not true easterly winds.

Thanks for bringing that to our attention, we have looked at the mean 800hPa zonal winds and the area is in the westerly belt (Fig. RR7). We've added (weaker westerly winds) to the revised manuscript. (Line 334 of the track changes revised manuscript).

[Figure]

*Fig. RR7: Composite maps of the winter ERA5 mean 800hPa zonal wind (shading) for the Mount Brown South site average (MBS Cl−) (a) upper [1982, 1986, 1987, 1993, 1996, 1997, 2009, 2012, 2014, 2015] and (c) lower quartile [1980, 1981, 1983, 1990, 1991, 1995, 1999, 2001, 2010, 2011] and the Niño3.4 (b) upper [1982, 1987, 1991, 1993, 1997, 2002, 2004, 2009, 2012, 2015] and (d) lower quartile [1984, 1985, 1988, 1989, 1998, 1999, 2000, 2007, 2010, 2011]. The MBS northeast coast box highlighted in fuchsia dashed line and the blue dot indicates MBS.*

Lines 275-277: This section should also mention blowing snow on sea ice as a major source of sea salt, in addition to frost flowers.

Thank you for the suggestion but we stand by what we wrote here in the original manuscript. Since we are specifically talking about newly formed sea ice and the limitation to using SIC, we do not think blowing snow needs to be mentioned as that can be over any age of sea ice.

Line 287: "The high MSLP anomaly above MBS…". Echoing a previous comment, I believe this means "north of MBS", rather than over MBS.

Thanks for bringing that to our attention, we've revised the manuscript accordingly. (Lines 352 and 354 of the track changes revised manuscript).

 Figure 7a does not support the idea that "El Nino is related to ….new sea ice formation", as there is essentially no SIC anomaly in the MBS sea ice source region associated with El Nino.

Thank you for your insightful comment. We agree that the positive sea ice concentration (SIC) anomalies during El Niño events in the green box are indeed very weak. We have noted this in the manuscript. (Line 338 of the track changes revised manuscript).

**RC3**

Shea et al. provide an analysis of possible mechanisms linking tropical sea-surface temperature variability (Nino 3.4) to Mount Brown South-regional sea ice concentration as recorded by the MBS ice core. This atmosphere-transmitted teleconnection between the tropical Pacific and Antarctica is well known, and is a process that would benefit from ice core paleoclimate reconstruction if indeed a strong enough modern proxy relationship can be demonstrated.

As the authors point out in section 5.1, the correlation is demonstrated but the mechanism linking to sea ice concentration is not made clear by this analysis. A broader analysis including other seasons or combinations of months would be beneficial. It is very important to understand proxy relationships for the MBS core, as that millennial length record is indeed in a region of very limited recent or paleoclimate information. I think the manuscript needs revisions given the concerns of reviewers 1 and 2, but after these considerations would be appropriate to publish in CP.

We thank the reviewer for their time and valuable input. Below, we detail the revisions made in response to their suggestions.

Main comments:

Many references to other MBS work (Jackson, Udy, Vance, Crockart) but less of broader evidence from the literature. What similarities do the West Antarctic ITASE cores have, with respect to the proposed mechanisms affecting sea salt concentrations (work of Dixon and others)? A deeper analysis of the literature would improve the manuscript.

Thank you for your valuable feedback. We recognize the importance of incorporating broader evidence from the literature to strengthen our manuscript. The West Antarctic ITASE cores share several similarities with respect to the proposed mechanisms affecting sea salt concentrations, as highlighted in the work of Kaspari and Dixon. These mechanisms include the influence of sea ice extent, the presence of open-water features, wind strength and direction, as well as the positioning of low-pressure systems, play significant roles in the transport and deposition of sea salt aerosols. We will conduct a deeper analysis of the literature to further elucidate these mechanisms and enhance the manuscript. (Lines 400 and 404 of the track changes revised manuscript).

There is a conflation of El Niño and ENSO throughout the paper that should be reconsidered. It would seem that if only SST correlations are examined, as they show the strongest significant

correlation, the authors need to limit their interpretations to El Niño albeit knowing that the teleconnection between tropical Pacific SST is transmitted through the atmosphere.

Thank you for your insightful suggestion. We have taken your feedback into account and made some revisions to address the conflation of El Niño and ENSO throughout the paper.

Firstly, we have included an analysis for the SOI to provide a more comprehensive view of the atmospheric component of ENSO (Table RR1 and Fig. RR6). Furthermore, we have enhanced our presentation of the MSLP maps by adding shading to better illustrate the atmospheric conditions associated with ENSO (Fig. RR6, Fig. A1). These maps highlight the pressure dipole in the tropics over Darwin and Tahiti. This pressure dipole is a key feature of the ENSO phenomenon, demonstrating the interconnectedness of the oceanic and atmospheric components.

In Fig. 1 and Fig. A1, the observed pressure dipole supports the presence of ENSO-related anomalies, reinforcing the validity of our approach. By including both SST and MSLP data, we provide a more integrated view of the ENSO dynamics, which justifies our interpretation and use of the term ENSO in the manuscript.

We believe these revisions address the concern regarding the conflation of El Niño and ENSO, and provide a clearer and more robust analysis of the teleconnections. We appreciate your feedback and believe that these changes enhance the clarity of our manuscript.

Line-by-line comments:

L24-25: none of the West Antarctic references use ice cores to understand ENSO effects. Following statement then about East Ant. lack of ice cores is unrelated to understanding ENSO linkages

This is a reasonable comment and this sentence has been removed from the revised manuscript. (Lines 27 and 30 of the track changes revised manuscript).

L28 change to "… links to drivers of Antarctic climate"

Thank you for your suggestion. We have already made the change as recommended by RC2.

L 48 reference to Abrams et al., 2013

The updated manuscript now includes this citation to support the relevant discussion. (Line 107 of the track changes revised manuscript).

L81 remove redundant "remove long term trends and"

That's a helpful suggestion, and we've corrected it in the manuscript. (Line 64 of the track changes revised manuscript).

L85 add a little more justification from the Vance 2024 reference to make clear why the annual sea salt conc is winter dominated as this is not obvious. Just in one additional sentence, perhaps.

Thank you for your feedback. We have already made the change as recommended by RC1.

L90 what months are taken for austral winter?

Thank you for your feedback. We have already made the change as recommended by RC1.

L111 only selecting two relatively anomalous years also risks them being outliers…

We have added all the years in the MBS upper and lower quartiles. (Line 133 of the track changes revised manuscript).

L242 are these "high precipitation storms" atmospheric rivers? I suspect so, in many cases and encourage looking into the work of Wille, Gorodetskaya and others on the subject

That is an interesting hypothesis but beyond the scope of this study

L307 Change "stem" to something like "originate"

Thanks for the suggestion, we've addressed this in the manuscript. (Line 375 of the track changes revised manuscript).

L325 It would have been useful to see a reanalysis-based examination of this MBS - Law Dome disagreement to see if mechanisms could be teased out. Perhaps beyond the scope of the paper but a lingering limitation as they indicate.

That would be interesting to analyse but unfortunately it is beyond the scope of this study

L 332 add more specifics about these additional studies (locations in more detail…) as they indeed provide good broader context.

Thank you for your suggestion. We agree that providing more specifics about the locations of these additional studies will enhance the broader context of our manuscript and have updated the manuscript to include this. (Line 398 of the track changes revised manuscript).

L 333 Drive this home more. These findings are comparable, so what are the commonalities across the sites? Is it generally similar that high SSC upwind of ice core sites during heavy snowfall events is conducive to greater SSC at ice core sites? Presumably so.

Thank you for your comment. We have emphasized this more in the manuscript.

L 340 indeed why do the authors not explore other seasons? This should be a relatively minor change to code to plot summer rather than winter.

RC1 and RC2 also commented on looking at spring which is now in the appendix section of the revised manuscript (Figs A3-A9) however we do not look at summer or autumn because the MBS annual sea salt concentration is winter dominated.

L 341 yes ENSO is not the same as El Niño and the authors should be more careful and consistent with their references to these phenomena. I'd argue it's not appropriate to discuss ENSO at all if only Nino indices are used.

We have addressed this from your main comment.

L345-347 seems like a superfluous statement and much work on flavors of Pacific SST and their influence on Antarctica has been done (work of Ding, Steig, others).

Thanks for the suggestion, we've revised this in the manuscript. (Line 423 of the track changes revised manuscript).

Figure 9 is a little too abstract… what is the red blob and blue blob generated from? Presumably correlation maps presented earlier, and if so that should be made clear in the figure caption. I do understand the diagram and it is actually fairly clear, but could perhaps be refined for clarity. Some authors choose to contract an artist or graphic designer for this work, which does have some cost but improves clarity.

We have added to the Figure 9 caption that the schematic is generated from the composite maps seen in Figs 2,3,5,6.

References

Kannan, K. S., Manoj, K., and Arumugam, S.: Labeling methods for identifying outliers, International Journal of Statistics and Systems, 10, 231–238, 2015.

Lu, Z. and Liu, Z.: Orbital modulation of ENSO seasonal phase locking, Climate Dynamics, 52, 4329–4350, 2019.

---

## Referee Report (RR1)

Attempts to identify the mechanism linking sea salt concentration to ENSO at MBS, building on Crockery et al. and Udy et al., both 2021. There is an El Niño but not a La Niña relationship, which is not particularly surprising given that the atmospheric response to El Niño has a relatively well understood mechanistic connection to Antarctic atmosphere-ocean anomalies. The opposite Pacific SST conditions don't force the same response, leaving other mechanisms to potentially dictate variability in Antarctic Ocean and sea ice conditions in those years.

The same uncertainty in precise mechanistic link between tropical Pacific variability remains in the manuscript, but is acknowledged more explicitly.

Line 44 remove period before parenthetical containing references.

Line 58 careful to remember that climate indices like Antarctic Dipole are merely statistical descriptions of variability in climate/environmental parameters.

Line 62 clarify that the Rossby wave is an atmospheric wave (just to not confuse reader between ocean or atmospheric "wave")

Line 120 there "are" known biases in ERA5… Good references for ERA5 anomalies also come from Bromwich et al. as recently as 2024 (GRL)

Line 127 read for consistent use of data in plural

---

## Author Response (AR2)

In the response document, reviewer comments I have responded to are shown in blue text In the response document, responses are shown in black text

Line numbers refer to the track changes revised manuscript

**RC1**

The authors have made substantial changes to the manuscript, addressing most of my initial comments. I only have a few minor observations to note.

\*All line references pertain to the non-track changes version of the manuscript.

**General minor comment:**

The manuscript presents the ERA5 zonal and meridional winds at 800 hPa. One of the primary uses of this wind dataset is to correlate wind anomalies with sea ice concentration, as mentioned in lines 233-237:

"These relationships can be understood physically: the correlations are consistent with the mean–state northerly winds transporting sea ice towards the coast and therefore, a weakening in these winds implies less transport towards the coast and thus increased SIC in the outer pack. The strengthening of the mean–state westerly winds will enhance northward Ekman transport of sea ice (increasing SIC), as Ekman transport theory dictates that sea ice will drift 45°to the left of the wind direction (UCAR, 2008; Purich et al., 2016), also increasing SIC in the outer ice pack."

Additionally, it is used to connect the wind vector with the transport of sea salts to the MBS site, as referenced in lines 244-246:

"This indicates that, for both high and low sea salt deposition conditions, on the days of highest precipitation the winds blow over the northeast coast box sea ice anomaly area, and then blow towards MBS, depositing sea salt aerosol at the site that has been scavenged by precipitation (i.e. wet deposition)."

The application of wind data is further summarized in section 3.3, lines 293-296:

"Firstly, strengthened westerly winds increase sea salt aerosol lofting from open ocean sea spray. Secondly, in response to the strengthened westerly winds, there will be enhanced northward Ekman transport of sea ice. This ice transport mechanism will increase the SIC on the outer (equatorward) side of the ice pack and increase sea ice formation closer to the coast, with sea ice providing a source of sea salt, either by blowinng snow over sea ice or frost flowers (Frey et al., 2020)."

As presented above, the ERA5 800 hPa data is utilized to interpret "surface" processes. However, the 800 hPa winds does not necessarily reflect surface wind variability accurately. An assessment of how 800 hPa winds relate to surface winds would help clarify whether the variability observed at 800 hPa is representative of surface wind conditions. This analysis could be included in the supplementary materials.

We thank the reviewer for their second review of our manuscript, and their thoughtful consideration of the difference between 800 hPa and surface winds. However, for the sake of the analysis in this manuscript, we feel the 800 hPa winds are appropriate to represent near-surface processes around the Antarctic margins.

The initial study examining the ENSO link to MBS was made using 10 metre surface winds (Crockart et al., 2021), as they were interested in the source regions of the sea salt, which is of course the surface ocean. In contrast, in this study we are looking at mechanisms relating to the larger scale (regional) circulation, including the circulation that is incident at the ice core site (~2000 metres asl). Using surface winds may give an incorrect result for the high-altitude continent (and ice core site), so we would prefer to stick with a level above the surface, as is commonly used in other studies to represent 'near' surface processes. Further, we are not looking at blowing snow transport (a surface process), but rather aerosol transport (near surface). Taken together, we consider that using surface winds would not reflect the full transport pathways well. Thus, we think it more reasonable to use 800 hPa data at these high latitudes and in this study.

**Specific minor comments:**

Line 24: I suggest adding a line introducing ice cores to smoothly transition from the importance of paleoclimate reconstructions to the importance of a well-dispersed network of ice cores.

Inserted sentence to link these two points. Lines 24-25.

Line 31: I suggest including the temporal timeframe used to obtain that correlation Added

Line 44: remove the full stop before the citation

Removed

Line 81: It is mentioned that four cores were drilled at MBS, but only three are used for the "site average". Could you include a short explanation of why there is a fourth core that was not used?

Added explanation that the fourth ice core was retained for persistent organic pollutant studies. Lines 83-84.

Line 118: add a space after "1979-2016"

Added

Line 131: I believe the reference should say "Bureau of Meteorology"

Fixed

Figure 2 caption: change (B) and (D) for lower-case

Fixed

Table 1 Caption: correct "Nino" for "Niño"

**Corrected**

Line 289: correct "La Nina" for "La Niña"

Corrected

Line 318: remove the double full-stop

Removed

Line 346: As the text mentions the influence of ENSO in Law Dome, it would be useful to have a reference of where Law Dome is located. Maybe in Figure 6 and/or Figure 8.

Thanks for this suggestion. We have added a marker for the location of Law Dome to Fig. 6.

**RC3**

Attempts to identify the mechanism linking sea salt concentration to ENSO at MBS, building on Crockery et al. and Udy et al., both 2021. There is an El Niño but not a La Niña relationship, which is not particularly surprising given that the atmospheric response to El Niño has a relatively well understood mechanistic connection to Antarctic atmosphere-ocean anomalies. The opposite Pacific SST conditions don't force the same response, leaving other mechanisms to potentially dictate variability in Antarctic Ocean and sea ice conditions in those years.

The same uncertainty in precise mechanistic link between tropical Pacific variability remains in the manuscript, but is acknowledged more explicitly.

We thank the reviewer for their second review of our manuscript, which we appreciate. We feel that the mechanistic link remains limited but agree that our manuscript acknowledges limitations clearly.

Line 44 remove period before parenthetical containing references.

Removed

Line 58 careful to remember that climate indices like Antarctic Dipole are merely statistical descriptions of variability in climate/environmental parameters.

We have added the phrase a 'simplified statistical description of' to the description of the Antarctic Dipole to make this clear. Line 61.

Line 62 clarify that the Rossby wave is an atmospheric wave (just to not confuse reader between ocean or atmospheric "wave")

Clarified

Line 120 there "are" known biases in ERA5... Good references for ERA5 anomalies also come from Bromwich et al. as recently as 2024 (GRL)

Changed is to are and added reference.

Line 127 read for consistent use of data in plural

Fixed for more consistent use of data in plural in the Data section.